# Generative power of a protein language model trained on multiple sequence alignments

**Damiano Sgarbossa[1,2], Umberto Lupo[1,2]\*, Anne-Florence Bitbol[1,2]\***

[1]Institute of Bioengineering, School of Life Sciences, École Polytechnique Fédérale de Lausanne (EPFL), Lausanne, Switzerland; [2]SIB Swiss Institute of Bioinformatics, Lausanne, Switzerland

**Abstract** Computational models starting from large ensembles of evolutionarily related protein sequences capture a representation of protein families and learn constraints associated to protein structure and function. They thus open the possibility for generating novel sequences belonging to protein families. Protein language models trained on multiple sequence alignments, such as MSA Transformer, are highly attractive candidates to this end. We propose and test an iterative method that directly employs the masked language modeling objective to generate sequences using MSA Transformer. We demonstrate that the resulting sequences score as well as natural sequences, for homology, coevolution, and structure-based measures. For large protein families, our synthetic sequences have similar or better properties compared to sequences generated by Potts models, including experimentally validated ones. Moreover, for small protein families, our generation method based on MSA Transformer outperforms Potts models. Our method also more accurately reproduces the higher-order statistics and the distribution of sequences in sequence space of natural data than Potts models. MSA Transformer is thus a strong candidate for protein sequence generation and protein design.

**\*For correspondence:**
umberto.lupo@epfl.ch (UL);
anne-florence.bitbol@epfl.ch
(A-FB)

**Competing interest:** The authors declare that no competing interests exist.

## Editor's evaluation

This important study proposes a method to sample novel sequences from a protein language model that could have exciting applications in protein sequence design. The claims are supported by a solid benchmarking of the designed sequences in terms of quality, novelty and diversity.

## Introduction

Designing new proteins with specific structure and function is a highly important goal of bioengineering. Indeed, it can allow to tune their stability or their biochemical properties, including their enzymatic activities, enabling important medical applications. The search for novel proteins is difficult due to the huge size of protein sequence space: for instance, there are $20^{100}$ different possible sequences for a short protein domain with 100 amino acids. Furthermore, only a small fraction of this space comprises sequences that do fold, as demonstrated by experiments studying random sequences (*Socolich et al., 2005*), and by theoretical arguments based on the physics of disordered systems (*Bialek, 2012*). De novo or rational protein design, which starts with target three-dimensional structures and physico-chemical potentials, can generate proteins which are not in a known protein family (*Dahiyat and Mayo, 1997*; *Kuhlman et al., 2003*; *Liang et al., 2009*), but is generally restricted to small proteins (*Rocklin et al., 2017*). Conversely, directed evolution allows to perform a local search of sequence space, but generally remains limited to the vicinity of a natural sequence (*Arnold, 2018*).

Generative computational models that build on the breadth of available natural protein sequence data, and capture a representation of protein families, now offer great alternatives that can allow to sample novel sequences belonging to protein families. In particular, Potts models, or DCA models (*Weigt et al., 2009*; *Morcos et al., 2011*; *Marks et al., 2011*; *Ekeberg et al., 2013*), which are pairwise maximum entropy models trained to reproduce the one- and two-body statistics of the sequences of a family, allow direct sampling from a probability distribution modeling this family (*Figliuzzi et al., 2018*), and have been used successfully for protein design (*Russ et al., 2020*). Variational autoencoders are deep learning models which also allow sampling, and they have been shown to successfully produce functional proteins (*Hawkins-Hooker et al., 2021a*), although their statistical properties appear to have a lower quality than with Potts models (*McGee et al., 2021*).

Protein language models are deep learning models based on natural language processing methods, especially attention (*Bahdanau et al., 2015*) and transformers (*Vaswani et al., 2017*). They are trained on large ensembles of protein sequences, and capture long-range dependencies within a protein sequence (*Alley et al., 2019*; *Elnaggar et al., 2021*; *Rives et al., 2021*; *Rao et al., 2021b*; *Vig et al., 2021*; *Madani et al., 2020*; *Madani et al., 2021*; *Rao et al., 2021a*). These pre-trained models are able to predict structure in an unsupervised way (*Rao et al., 2021b*), either taking as input a single sequence (*Rives et al., 2021*) or a multiple sequence alignment (MSA) (*Rao et al., 2021a*), potentially by transferring knowledge from their large training set (*Bhattacharya et al., 2020*; *Bhattacharya et al., 2022*). The great success of supervised protein structure prediction by AlphaFold (*Jumper et al., 2021*) is partly based on the use of transformers. It is therefore of strong interest to assess the generative ability of protein language models, and recent works show that this has high potential (*Madani et al., 2021*; *Johnson et al., 2021*; *Hawkins-Hooker et al., 2021b*; *Ferruz et al., 2022*; *Hie et al., 2022*).

Correlations in amino-acid usage that can be observed between the columns of MSAs of homologous proteins (*Casari et al., 1995*; *Lapedes et al., 1999*; *Dunn et al., 2008*) were experimentally demonstrated to be highly important to generate functional synthetic proteins (*Socolich et al., 2005*; *Bialek and Ranganathan, 2007*). The importance of pairwise coevolution signals was then corroborated by the success of Potts models at predicting structural contacts (*Weigt et al., 2009*; *Marks et al., 2011*; *Morcos et al., 2011*; *Sułkowska et al., 2012*; *Ekeberg et al., 2013*), analyzing mutational effects (*Dwyer et al., 2013*; *Cheng et al., 2014*; *Cheng et al., 2016*; *Figliuzzi et al., 2016*), protein evolution (*de la Paz et al., 2020*) and conformational changes (*Morcos et al., 2013*; *Malinverni et al., 2015*), designing proteins (*Russ et al., 2020*), and predicting protein–protein interaction partners (*Bitbol et al., 2016*; *Gueudré et al., 2016*; *Cong et al., 2019*; *Green et al., 2021*). Protein language models that take MSAs as input (*Rao et al., 2021a*; *Jumper et al., 2021*) are able to directly exploit this covariation signal, and are thus particularly interesting candidates for protein design. Thus motivated, we focus on MSA Transformer (*Rao et al., 2021a*), a protein language model which was trained on MSAs using the masked language modeling (MLM) objective, without additional supervised training – by contrast to AlphaFold (*Jumper et al., 2021*). We ask how the generative properties of MSA Transformer compare to those of Boltzmann machine DCA (bmDCA) (*Figliuzzi et al., 2018*; *Russ et al., 2020*), a state-of-the-art generative Potts model.

We propose and test a generating method that directly uses the MLM objective in an iterative way to generate sequences using MSA Transformer. Using homology, coevolution, and structural scores, we demonstrate that the sequences generated by this method score as well as natural sequences. We further show that this good performance is not restricted to synthetic sequences that are very similar to natural sequences. For large protein families, our synthetic sequences have homology and structure-based scores as good as or better than sequences generated by bmDCA, and have similar properties to experimentally validated bmDCA-generated sequences. Moreover, for small protein families, our generation method based on MSA Transformer outperforms bmDCA, by providing synthetic sequences that score well without being extremely similar to natural ones. However, we find that bmDCA better reproduces the one- and two-body statistics of the natural MSAs than MSA Transformer when used with default parameters, consistently with its training objective. Interestingly, the opposite generally holds for higher-order statistics. MSA-Transformer–generated sequences also better reproduce the distribution of sequences in sequence space than bmDCA-generated ones. Our conclusion is that MSA Transformer is a strong candidate for protein sequence generation and protein design.

# Results

## An iterative masking procedure allows MSA Transformer to generate novel sequences with high scores

Can the protein language model MSA Transformer (*Rao et al., 2021a*) be used to generate sequences that are credible members of protein families? How do its generative abilities compare to Potts models inferred by bmDCA (*Figliuzzi et al., 2018*), a state-of-the-art generative DCA method which has been experimentally shown to generate functional proteins (*Russ et al., 2020*)? To address these questions, we developed and employed an iterative masking procedure to generate synthetic MSAs from natural MSAs of 14 different large Pfam protein families (see *Appendix 1—table 5*) and 7 small ones (see *Appendix 1—table 6*) with MSA Transformer, as described in 'Using MSA Transformer to generate sequences via an iterative masking procedure'. We also generated synthetic sequences by Markov Chain Monte Carlo (MCMC) sampling from Potts models inferred from these MSAs by bmDCA, using two variants that differ by sampling temperature $T$ and regularization strength $\lambda$, matching, respectively, the default parameters employed in *Figliuzzi et al., 2018*, and some of those used in *Russ et al., 2020*, see 'Sampling sequences from Potts models' for details. For each protein family, we thus obtained four different MSAs of the same depth: the natural one, the one generated by our iterative masking procedure using MSA Transformer, and the two MSAs sampled from the inferred Potts model. To characterize each sequence, we consider four different scores (see 'Scoring individual sequences'). First, we assess the quality of the generated sequences as homologs of the protein family of interest; we do this via the HMMER (http://hmmer.org) score of the hidden Markov model employed by Pfam to retrieve natural homologs. Second, we consider a score that accounts for coevolution between amino-acid sites, namely the statistical energy score from the Potts model fitted on the natural MSA. Third, we determine AlphaFold's confidence in its determination of the three-dimensional structure of these sequences, via the predicted local-distance difference test (pLDDT) score. Fourth, to assess whether the predicted structures are similar to the native ones, we compute the root-mean-squared deviation (RMSD) between a reference experimental structure and the Alpha-Fold predicted structures. The first three scores are such that higher values are better, while smaller RMSD values indicate that predicted structures are similar to the native ones. Together, these scores account for very different aspects of proteins, namely homology, coevolution and structure.

Let us first consider the 14 large protein families in *Appendix 1—table 5*, where MSAs are deep enough to accurately fit Potts models using bmDCA (*Figliuzzi et al., 2018*). *Figure 1* shows that, for all these protein families, and for these four different scores, the sequences generated by MSA Transformer using our iterative masking procedure have scores that are at least as good as those of natural sequences, and better than those of sequences generated by bmDCA with default parameters (*Figliuzzi et al., 2018*), as confirmed by the Kolmogorov–Smirnov test (see *Appendix 1—table 1*). Decreasing the sampling temperature and the regularization strength used with bmDCA improves the statistical energy score as expected (*Russ et al., 2020*), but also other scores. These other scores, and most importantly our two structural scores, are similar or better for MSA-Transformer–generated sequences compared to those generated by bmDCA with non-default parameters. In particular, the median pLDDT score is larger for the former than for the latter in 11 protein families out of 14, by a margin larger than the standard deviation in 4 of them (see *Appendix 1—table 2*). These results demonstrate that MSA Transformer is a good candidate to generate synthetic sequences from protein families, and that our iterative masking procedure allows to perform such generation.

How different are these synthetic sequences from the natural ones? In particular, are the best-scoring sequences novel, or are they almost copies of natural sequences? In *Figure 2* we show, for two example protein families (PF00072 and PF00153), the HMMER score and the DCA statistical energy score versus the sequence's Hamming distance to its closest natural sequence in the natural MSA.

From the marginal distributions of the Hamming distances in *Figure 2*, we observe that MSA Transformer generates sequences with variable distances to their closest natural sequences, and that these distances are overall larger than those between natural sequences and their closest neighbors (excluding themselves). With default parameters, bmDCA generates sequences which are generally very different from the natural ones, but decreasing sampling temperature makes bmDCA-generated sequences more similar to natural ones and to each other, see *Figure 1—figure supplement 1*. Besides, the marginal distributions of scores illustrate the general observation made on *Figure 1* and in *Appendix 1—table 2* that MSA-Transformer–generated sequences have good scores. Moreover,

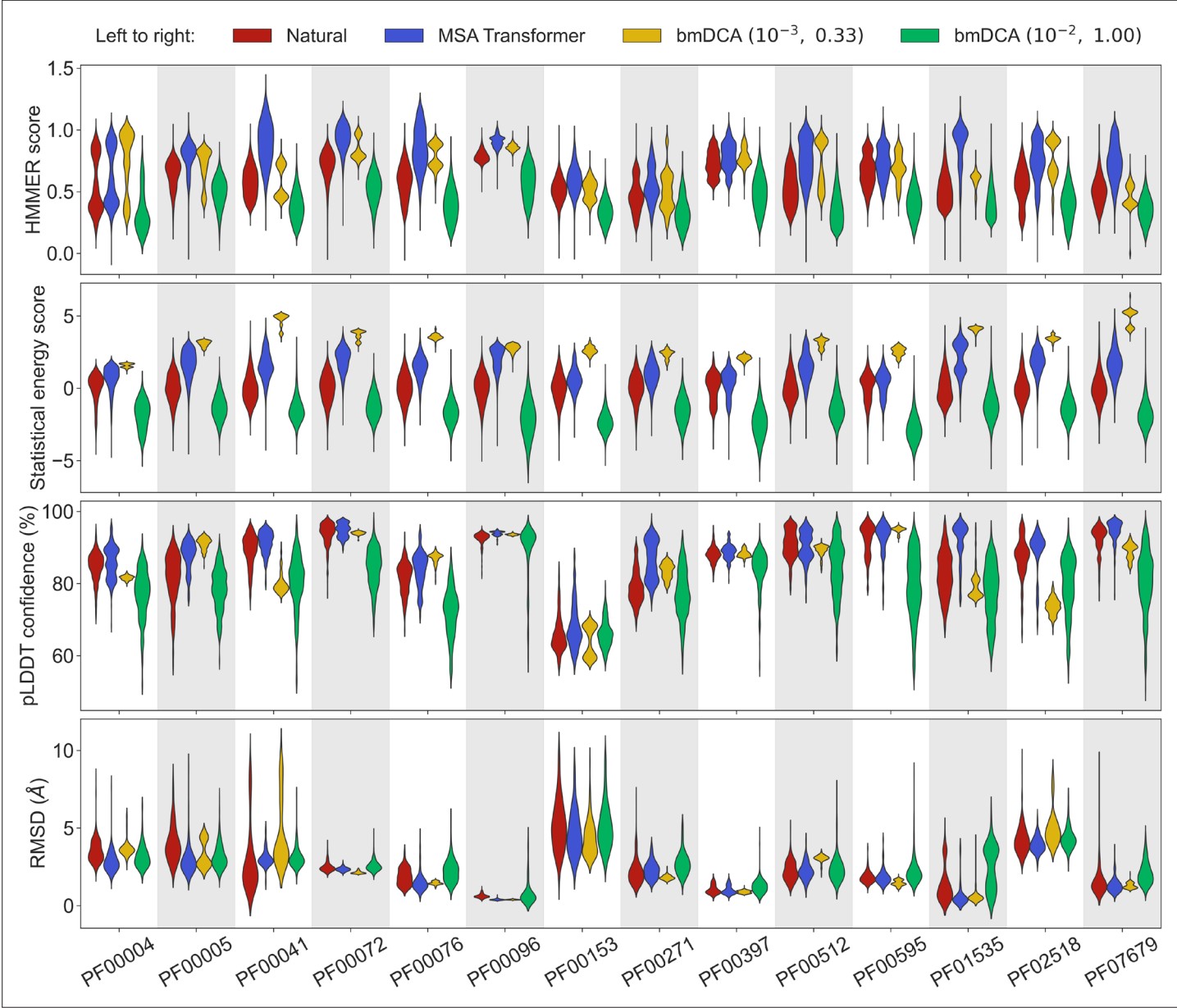

**Figure 1.** Comparison of homology, coevolution, and structure-based scores between natural sequences and sequences generated by MSA Transformer or Boltzmann machine DCA (bmDCA). For each Pfam family in *Appendix 1—table 5*, we compare a natural MSA from Pfam and three synthetic MSAs of the same depth. The first synthetic MSA was obtained using MSA Transformer via our iterative masking procedure, and the second and third ones were generated by a Potts model inferred from the natural MSA using bmDCA with two different pairs $(\lambda, T)$ of regularization strength $\lambda$ and sampling temperature $T$. For each of the four scores described in 'Scoring individual sequences', we show the distributions of score values among sequences in each MSA as a violin plot. Higher score values are better for all scores except root-mean-squared deviation (RMSD) (bottom panel), where smaller values indicate a closer match to an experimental structure. Top panel: For each Pfam family, HMMER scores are divided by the highest score found in the natural MSA. Note that sequences below HMMER's default homology detection score ($E$-value larger than 10), and whose HMMER score is thus 0, are not shown (the median over families of the fraction of such sequences is 2% for bmDCA ($10^{-2}$, 1.00)-generated MSAs, while there are no such sequences among the MSA-Transformer–generated ones). Second panel: Statistical energy scores are defined as minus the bmDCA statistical energies. To accommodate the highly family-dependent ranges of these scores, for each Pfam family we show their values after shifting by the mean score in the natural MSA, and normalizing by the standard deviation of natural MSA scores. Third panel: AlphaFold's predicted local-distance difference test (pLDDT) confidence scores. Bottom panel: RMSD of predicted structures with respect to the experimental structures in *Appendix 1—table 5*. Structural scores (pLDDT and RMSD) were computed on 200 randomly chosen sequences from each MSA. All kernel-smoothed histograms are normalized such that all violins have the same maximal width. Outliers (less than 1% in all cases) were discarded for legibility.

The online version of this article includes the following figure supplement(s) for figure 1:

**Figure supplement 1.** Multiple sequence alignment (MSA) diversity for each protein family and each generation method.

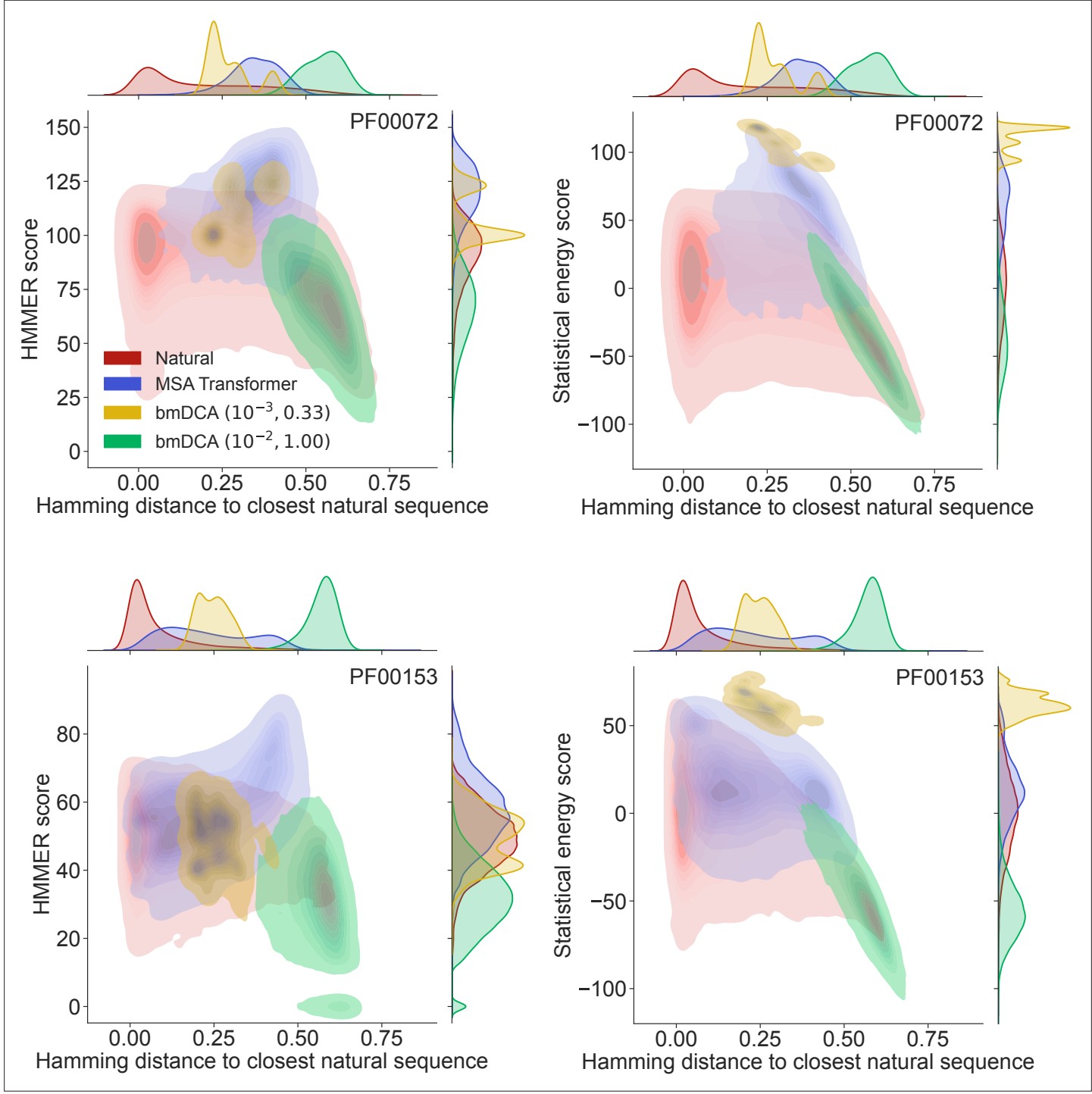

**Figure 2.** Homology and coevolution scores versus distance to the natural multiple sequence alignment (MSA), for protein families PF00072 and PF00153. We show contour plots of the HMMER score and the statistical energy score (defined as minus the DCA statistical energy, shifted by its mean value in the natural MSA) versus the Hamming distance of each sequence to the closest natural sequence (which is not itself, in the case of natural sequences). Results are shown for natural sequences and for sequences generated using MSA Transformer and Boltzmann machine DCA (bmDCA) (the same two $(\lambda, T)$ pairs as in *Figure 1* are used for bmDCA). The lightest contours shown include 99% of the cumulative probability mass.

the plots in *Figure 2* reveal that the MSA-Transformer–generated sequences featuring the highest HMMER scores tend to have large Hamming distances to natural sequences, that is to be truly novel (see also 'Choosing parameters in the iterative masking procedure'). We observe these trends for most large protein families studied, and they are robust to using BLOSUM similarity scores (*Henikoff and*

*Henikoff, 1992*) instead of Hamming distances. Therefore, our sequence generation method based on MSA Transformer is not reaching good scores by just reproducing natural sequences. Besides, the diversity of MSA-Transformer–generated MSAs, as measured by their effective depth (*Equation 8*), is only slightly smaller than that of natural MSAs (see *Figure 1—figure supplement 1*). Conversely, bmDCA at low temperature produces highly redundant sequences (*Figure 1—figure supplement 1*), which are concentrated in specific regions of the score versus distance space in *Figure 2*. Indeed, sequence generation by bmDCA is then constrained to exploring the local minima of the Potts model energy landscapes.

## Sequence generation by the iterative masking procedure is successful for small protein families

Accurately fitting Potts models requires deep and diverse MSAs, as evidenced by the strong dependence of structural contact prediction by Potts models on MSA depth (*Marks et al., 2011*; *Morcos et al., 2011*). By contrast, MSA Transformer was trained on many MSAs, and is able to transfer knowledge across protein families. It outperforms Potts models at unsupervised contact prediction most strongly for shallow MSAs (*Rao et al., 2021a*). How does sequence generation using our iterative masking procedure based on MSA Transformer compare to bmDCA in the case of small protein families?

To address this question, we generated synthetic MSAs starting from seven small families, using both our iterative masking procedure based on MSA Transformer and bmDCA with default parameters and with low sampling temperature. *Figure 3* reports all four scores discussed above in the case of these seven small families, listed in *Appendix 1—table 6*. We observe that MSA-Transformer–generated sequences have similar HMMER scores and structural scores to natural sequences. MSA-Transformer–generated sequences also generally have better HMMER scores and structural scores than those generated by bmDCA with default parameters. While low-temperature bmDCA yields better statistical energy scores (as expected), and also gives HMMER scores and structural scores comparable to natural sequences, it in fact generates sequences that are almost exact copies of natural ones (see *Figure 3*, bottom row). By contrast, MSA Transformer produces sequences that are quite different from natural ones, and have very good scores. Thus, our method based on MSA Transformer is particularly promising in the tricky case of small protein families.

## Higher-order statistics are better reproduced by MSA Transformer, while lower-order statistics are better reproduced by bmDCA

How well do synthetic MSAs generated by our method based on MSA Transformer, and by bmDCA, reproduce the statistics of amino-acid usage observed in natural MSAs? To address this question, we consider the r20 score (*Haldane et al., 2018*; *McGee et al., 2021*), which quantifies the statistical similarity of two datasets at various orders (see 'Analyzing the statistics of MSAs'). We compute it between each of our synthetic MSAs and the corresponding natural one, for the 14 large protein families in *Appendix 1—table 5*. We also present as reference an assumption-free null model, namely the r20 score between two subsets of each natural MSA. *Figure 4* shows that bmDCA with default parameters is most often the best method at reproducing lower-order statistics, while MSA Transformer is the best at reproducing higher-order statistics, in all families considered. bmDCA at lower temperature performs more poorly at reproducing the statistics of natural MSAs than other methods, because low-temperature biases the sampling (bmDCA models are effectively learned at temperature $T = 1$).

To have a more detailed insight into lower-order correlations, we estimate frequencies and information theory measures, at the one-, two-, and three-body level, from our natural and synthetic MSAs, and compare them (see 'Analyzing the statistics of MSAs'). *Figure 4—figure supplement 1* shows that one- and two-body statistics are generally better reproduced by bmDCA with default parameters than by MSA Transformer, while results are more mixed for three-body statistics. *Figure 4—figure supplement 2* and *Figure 4—figure supplement 3* show a comparison of second- and third-order connected correlations for PF00072 and PF00153. For PF00072, bmDCA reproduces better the second- but also third-order connected correlations of the natural data than MSA Transformer, while for PF00153, MSA Transformer reproduces the third-order connected correlations better than bmDCA, consistently with *Figure 4*. Potts models are pairwise maximum entropy models constrained to match the one- and two-body frequencies from natural MSAs. Thus, bmDCA is trained to reproduce these frequencies,

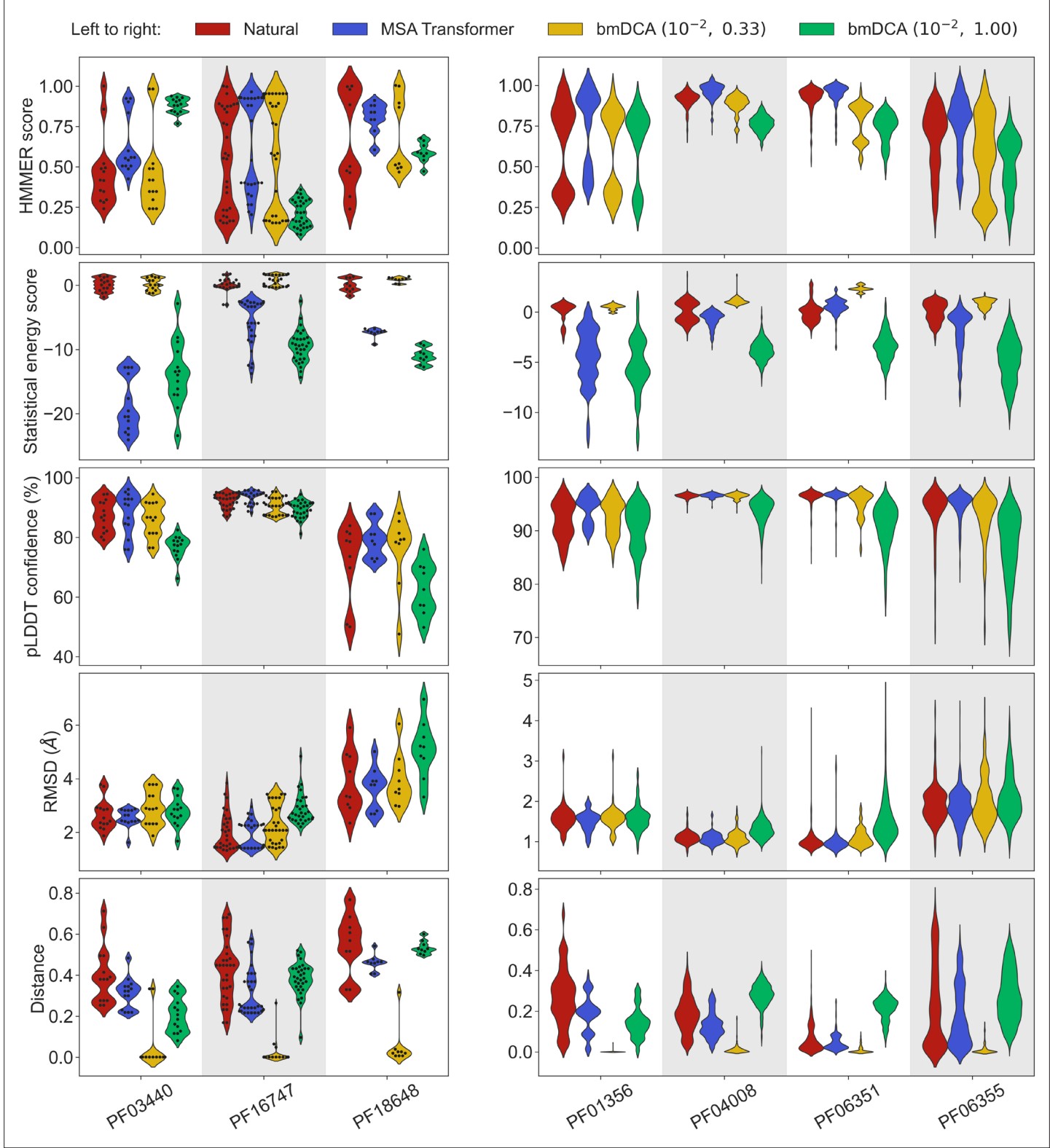

**Figure 3.** Application of our sequence generation method based on MSA Transformer to small protein families. We consider seven small protein families, with natural MSAs that comprise from nine to a few hundreds of sequences, see *Appendix 1—table 6*. As in *Figure 1*, for each family, we compare the natural MSA and three synthetic MSAs of the same depth. In all cases, we show violin plots of the same four scores as for large families in *Figure 1*, as well as of the Hamming distance to the closest natural sequence, which is not itself in the case of natural sequences ('Distance'). For the three smallest families (left panel; fewer than 40 sequences), we also show the score of each individual sequence as a swarm plot. Note that while

*Figure 3 continued on next page*

*Figure 3 continued*

we employ the same sampling temperatures $T$ as in *Figure 1* for Boltzmann machine DCA (bmDCA), here, we use regularization strength $\lambda = 10^{-2}$ throughout, due to MSA shallowness (see 'Sampling sequences from Potts models').

and achieves these objectives quite well, although the comparison to the null model in *Figure 4—figure supplement 2* and *Figure 4—figure supplement 3* hints that further improvements remain possible, see *Meshulam et al., 2021*. MSA Transformer has entirely different training objectives, but, interestingly, it performs comparably at reproducing three-body statistics and is better at reproducing even higher-order statistics than bmDCA.

## MSA Transformer captures well the distribution of sequences in sequence space

How are synthetic MSAs generated by MSA Transformer and bmDCA impacted by the heterogeneous repartition of natural sequences in sequence space? While natural protein sequences in a family

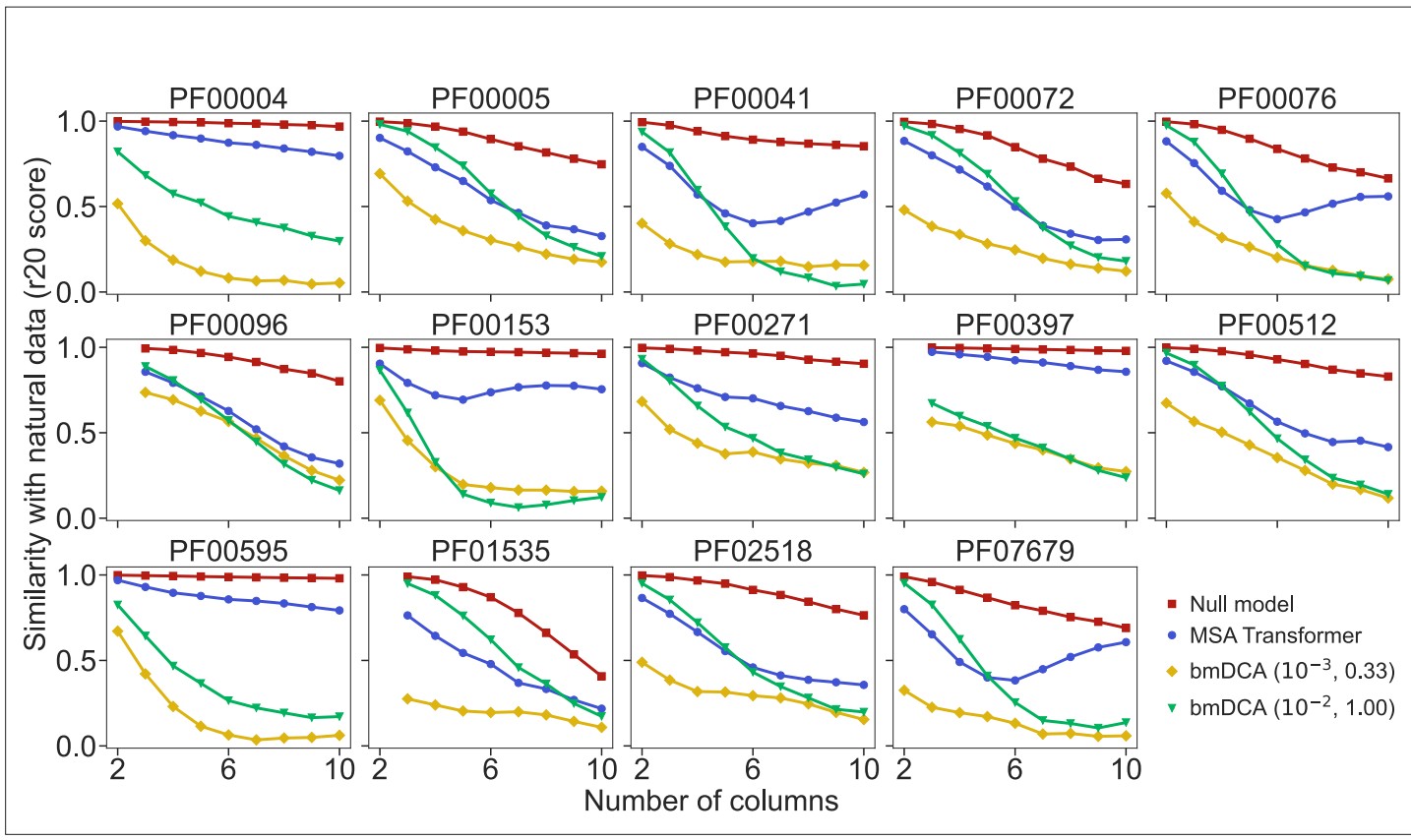

**Figure 4.** Similarity of statistics between synthetic and natural multiple sequence alignments (MSAs). To compare the statistics of synthetic and natural MSAs at various orders, we compute r20 scores (*Haldane et al., 2018*; *McGee et al., 2021*), and plot them versus the number of different MSA columns that are considered (see 'Analyzing the statistics of MSAs' for details). All families in *Figure 5* are considered. For each of them, the reference MSA comprises either half of the natural MSA (with sequences selected uniformly at random), or 30,000 sequences from it if the natural MSA depth is larger than 60,000. The null model compares the other half of the natural MSA to this reference MSA. It yields an estimate of the expected r20 scores due only to finite-size effects in a model-free, purely data-driven way.

The online version of this article includes the following figure supplement(s) for figure 4:

**Figure supplement 1.** Ability of generated sequences to reproduce one-, two-, and three-body statistics.

**Figure supplement 2.** Two- and three-body connected correlations estimated from generated multiple sequence alignments (MSAs) versus the natural one, for family PF00072.

**Figure supplement 3.** Two- and three-body connected correlations estimated from generated multiple sequence alignments (MSAs) versus the natural one, for family PF00153.

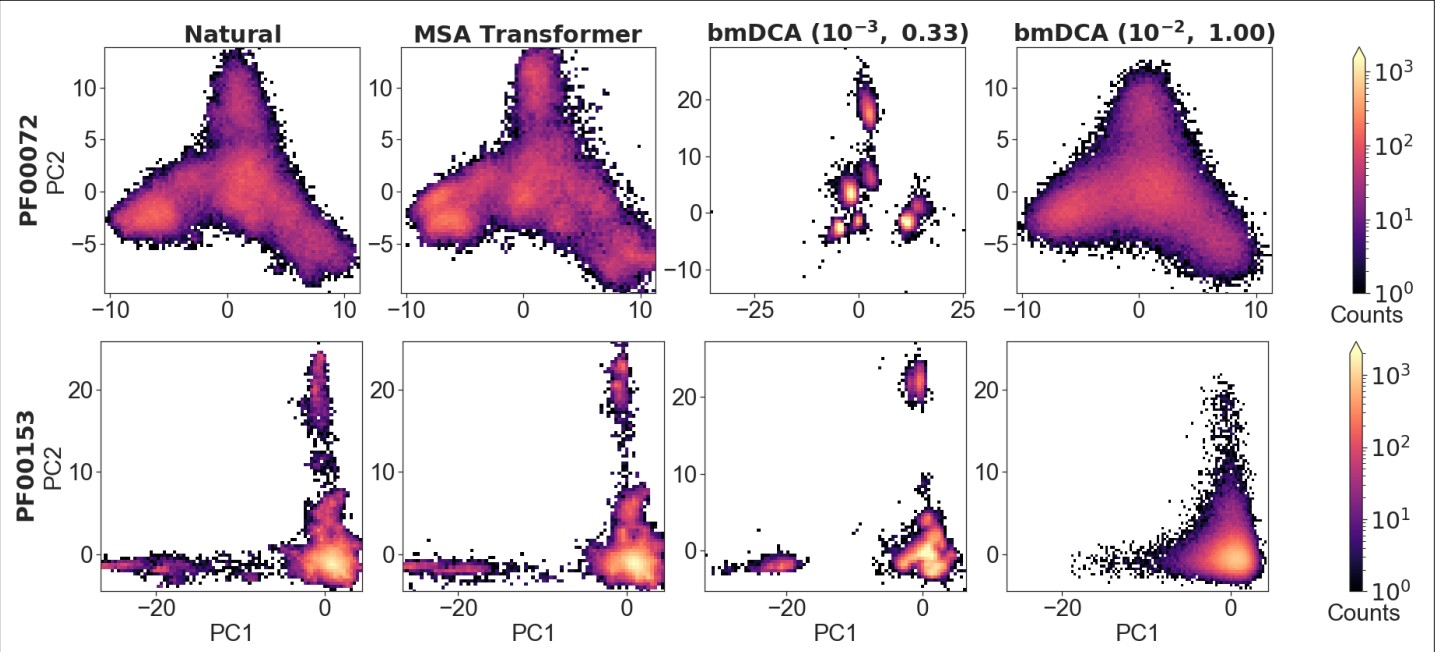

**Figure 5.** Distribution of sequences in sequence space, for families PF00072 and PF00153. We show the distribution of one-hot encoded natural and synthetic sequences projected in the subspace of the first two principal components of the natural multiple sequence alignment (MSA). The same axis limits are used within one family, except for Boltzmann machine DCA (bmDCA) ($10^{-3}$, 0.33) in the case of PF00072. Note that the fraction of the total variance explained by the first two principal components of each MSA is less than 4% for all families and all generation methods.

The online version of this article includes the following figure supplement(s) for figure 5:

**Figure supplement 1.** Distribution of sequences in sequence space for all large protein families in our dataset (part 1).

**Figure supplement 2.** Distribution of sequences in sequence space for all large protein families in our dataset (part 2).

**Figure supplement 3.** Neighbors of natural and synthetic sequences, for families PF00072 and PF00153.

**Figure supplement 4.** Comparing phylogenies inferred from natural and generated multiple sequence alignments (MSAs) for families PF00072 and PF00153.

have evolved from a common ancestor along a phylogeny, synthetic sequences do not have a real evolutionary history. However, as bmDCA and MSA Transformer are trained on natural data, they can capture phylogenetic correlations (*Lupo et al., 2022*). Besides, inferred Potts models are known to be impacted by phylogenetic correlations (*Weigt et al., 2009*; *Marks et al., 2011*; *Qin and Colwell, 2018*; *Vorberg et al., 2018*; *Rodriguez Horta et al., 2019*; *Rodriguez Horta and Weigt, 2021*; *Marmier et al., 2019*; *Colavin et al., 2022*; *Gerardos et al., 2022*; *Dietler et al., 2023*).

To analyze the overall distribution of MSA sequences in sequence space, we first perform a principal component (PC) analysis of one-hot encoded MSAs, and focus on the top two PCs (*Figliuzzi et al., 2018*) (see 'Characterizing the distribution of sequences in MSAs'). *Figure 5* shows the distribution of sequences in the space spanned by these top two PCs, for natural and synthetic MSAs, in the cases of PF00072 and PF00153. We observe that MSA Transformer is able to generate sequences with a distribution in sequence space that is very similar to that of the natural MSA. Conversely, bmDCA captures the overall shape of this distribution, but appears to smooth it compared to the natural data with default parameters and to restrict to sparse regions of the sequence space at low temperature, consistently with our previous results. These observations are general across all the deep MSAs we considered (see *Figure 5—figure supplement 1* and *Figure 5—figure supplement 2*). Note that a limitation of this analysis is that the top two PCs explain a small fraction of the variance in all cases (see *Figure 5*).

Next, to assess whether generated sequences most resemble natural ones that are well represented in their family or, rather, rare ones, we consider the closest natural sequence to each synthetic sequence, and count the neighbors of this natural sequence in the natural MSA (see 'Characterizing the distribution of sequences in MSAs'). *Figure 5—figure supplement 3* compares the distribution

of these numbers of neighbors for natural sequences and for the closest natural sequences to generated sequences, in the cases of PF00072 and PF00153. It shows that bmDCA generates sequences similar to natural sequences with fewer neighbors than typical in the natural data. Conversely, MSA Transformer generates sequences whose closest natural sequences have a distribution of number of neighbors similar to that of the natural MSA. This suggests that our generation method based on MSA Transformer tends to sample from denser regions of the sequence space than bmDCA, while not reproducing natural sequences (see also *Figure 2* and 'Choosing parameters in the iterative masking procedure').

Finally, to analyze in more detail the apparent relatedness of generated sequences, and compare it to real phylogenetic relationships in natural sequences, we infer phylogenetic trees from each synthetic and natural MSA, and analyze the eigenvalue spectrum of their modified graph Laplacian (MGL) to compare them (*Lewitus and Morlon, 2016*) (see 'Characterizing the distribution of sequences in MSAs'). *Figure 5—figure supplement 4* compares the density of these eigenvalue spectra for natural and synthetic MSAs regarding families PF00072 and PF00153. The skewness and the position of such distributions are indicators of the topology of the tree. In particular, distributions with negative skewness (right unbalanced) or which are shifted to the right, correspond to 'tippy' trees, while the opposite case corresponds to 'stemmy' trees (*Lewitus and Morlon, 2016*), which feature an accumulation of recent speciation events (short leaves length) (*Molina-Venegas, 2021*). In this light, *Figure 5—figure supplement 4* shows that both MSA Transformer and low-temperature bmDCA generate sequences with an apparent phylogeny that is more stemmy than the natural one, while bmDCA with default parameters yields a slightly more tippy tree. This is consistent with our observations regarding sequence diversity, which is larger than in natural data for bmDCA with default parameters, slightly smaller than in natural data using MSA Transformer and much lower using low-temperature bmDCA (see *Figure 1—figure supplement 1*).

## Comparison with published experimental datasets

How do the sequences generated by our method based on MSA Transformer compare to published protein design experimental datasets? Recently, sequences sampled from a bmDCA Potts model of the chorismate mutase protein family were experimentally demonstrated to be functional (*Russ et al., 2020*). In *Figure 6—figure supplement 1*, we show plots analogous to those in *Figure 2*, plus additional ones for our two structural scores (pLDDT and RMSD), in the case of chorismate mutase. This allows a detailed comparison between the sequences we generate using MSA Transformer and the sequences designed in *Russ et al., 2020* using bmDCA with a combination of different temperatures and regularization strengths. We find that our method based on MSA Transformer produces sequences that score as well as artificial sequences which have been tested experimentally. Besides, we obtained these results without fine-tuning the parameters of our generative procedure to this family, while several specific combinations of parameters were used in *Russ et al., 2020*.

To further compare our generated sequences to those tested experimentally in *Russ et al., 2020*, we consider relative enrichment, which is the experimental score used in *Russ et al., 2020* to assess the function of chorismate mutase enzymes. This score was measured in *Russ et al., 2020* for all sequences in the natural MSA and for sequences generated with bmDCA. We estimate the expected relative enrichment of our generated sequences as the relative enrichment of the closest natural sequence. To test our estimation procedure, we estimate the relative enrichments of the bmDCA-generated sequences from *Russ et al., 2020*, and we compare them to the experimentally measured values. We focus on the top third of sequences in terms of pLDDT scores, as it was shown in *Malbranke et al., 2021* that good structural scores help to select functional sequences. *Figure 6* shows that in this ensemble, sequences with a high (resp. low) estimated score have a high (resp. low) experimental score too. Next, we compare the distributions of estimated relative enrichment for sequences generated using our method based on MSA Transformer and for the bmDCA-generated sequences from *Russ et al., 2020*. *Figure 6* shows that they are quite similar to each other. This holds both when focusing on the top third of sequences in terms of pLDDT scores for each generation method, and when considering all generated sequences. Furthermore, in the high-pLDDT case, these distributions are quite similar to the distribution of measured relative enrichment for bmDCA-generated sequences. Importantly, a similar fraction of MSA-Transformer–generated sequences and of bmDCA-generated sequences have a large

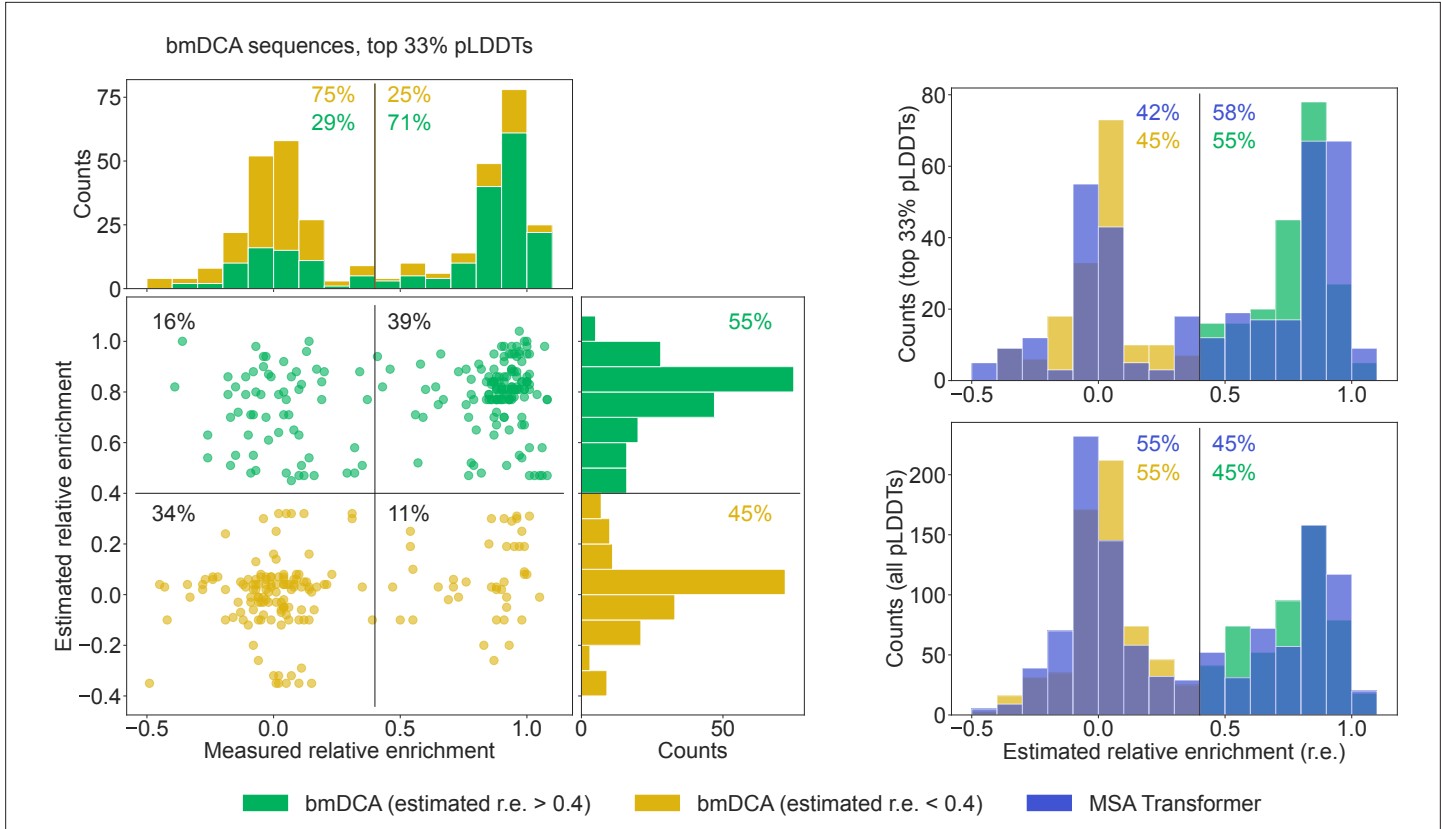

**Figure 6.** Comparison of our generated sequences to those experimentally tested in *Russ et al., 2020*, for the chorismate mutase family. Left: The estimated relative enrichment (r.e.) scores of the Boltzmann machine DCA (bmDCA)-generated sequences that are in the top 33% in terms of predicted local-distance difference test (pLDDT) scores are plotted versus their experimentally measured counterparts from *Russ et al., 2020*. We estimate the expected r.e. of these generated sequences as the r.e. of the closest natural sequence measured in *Russ et al., 2020*. We observe that high estimated r.e. is associated with high measured r.e., as 71% of sequences with estimated r.e. > 0.4 (green) also have measured r.e. > 0.4. Note that in the top marginals (showing the measured r.e. for bmDCA-generated sequences), the green and yellow histograms are stacked on top of each other. Thus, the stacked histogram shows the distribution of all measured r.e. values for bmDCA-generated sequences that are in the top 33% in terms of pLDDT scores. Top right: Overlaid histograms of estimated r.e. are shown for our MSA-Transformer–generated sequences and for the bmDCA-generated ones from *Russ et al., 2020*, restricting in both cases to the sequences with top 33% pLDDT scores. Bottom right: Same as top right, but considering all generated sequences.

The online version of this article includes the following figure supplement(s) for figure 6:

**Figure supplement 1.** Homology, coevolution, and structural scores versus distance to the natural multiple sequence alignment (MSA), for the chorismate mutase family.

**Figure supplement 2.** Deep mutational scanning (DMS) scores for families PF00595 and PF13354.

estimated relative enrichment. This suggests that our method based on MSA Transformer should be able to generate functional sequences.

While the data from *Russ et al., 2020* is particularly well suited to retrospectively evaluate our sequence generation method, we also propose a comparison of the distributions of scores based on experimental deep mutational scans (DMS) for protein families PF00595 (*McLaughlin et al., 2012*) and PF13354 (*Stiffler et al., 2015*). We compute these DMS scores for each natural and synthetic sequence, by summing the experimentally measured effects of the relevant single-point mutations with respect to the reference sequence of the experimental studies. *Figure 6—figure supplement 2* shows the distribution of the DMS scores of natural and generated sequences for these two families. Our sequence generation method based on MSA Transformer better reproduces the DMS score distribution of natural sequences than bmDCA, and generates sequences with better average DMS scores. Despite the potential limitations of our DMS scores, for example their additivity, these results corroborate our other findings and provide further encouragement for our sequence generation method based on MSA Transformer.

## Discussion

In this work, we proposed an iterative masking procedure which directly exploits the MLM objective of protein language models to generate sequences using the MSA-based neural language model MSA Transformer. We found that these sequences score as well as natural ones on three very different aspects, namely homology, coevolution, and structure. For large protein families, our synthetic sequences have homology and structure-based scores at least as good as bmDCA-generated sequences, and have similar properties to experimentally validated ones. Moreover, our generation method based on MSA Transformer is less limited by shallow MSAs than bmDCA, and is thus particularly promising for small protein families. Besides, MSA-Transformer–generated sequences better reproduce the higher-order statistics and the distribution of sequences in sequence space of natural data than bmDCA-generated ones. Conversely, bmDCA, with default parameters, better reproduces first- and second-order statistics, consistently with its training objective.

Our results are highly promising for sequence generation by MSA-based protein language models, and we hope that they will motivate further studies, especially experimental tests. They also show that protein deep learning models based on the MLM objective have great generative potential, despite not being obvious generative models. More generally, our results reinforce the new promising 'coevolution-driven' protein design approach of learning from sequences of evolutionarily related proteins the constraints associated to protein structure and function. This concept differs from structure- and physics-based de novo design (*Dahiyat and Mayo, 1997*; *Kuhlman et al., 2003*; *Liang et al., 2009*), and from the new possibility to use supervised deep learning models able to accurately predict protein structures (*Jumper et al., 2021*; *Baek et al., 2021*; *Chowdhury et al., 2023*) for structure-driven sequence generation (*Anishchenko et al., 2021*). One can view the coevolution-driven approach as intermediate between structure-based approaches and directed evolution ones (*Arnold, 2018*). The coevolution-driven approach was recently experimentally validated in the case of bmDCA Potts models, which capture pairwise coevolution patterns in MSAs (*Russ et al., 2020*), and for variational autoencoders (*Hawkins-Hooker et al., 2021a*; *McGee et al., 2021*). Protein language models trained on MSAs provide state-of-the-art unsupervised contact prediction and are able to capture coevolutionary patterns in their tied row attentions (*Rao et al., 2021a*), and capture phylogenetic relationships in column attentions (*Lupo et al., 2022*). This makes them ideal candidates to generate new protein sequences from given families. However, contrary to Potts models and variational autoencoders (*McGee et al., 2021*), they do not allow direct sampling from a probability distribution over sequences (*Goyal et al., 2021*). Here, we demonstrated the power of a simple generation method directly based on the MLM objective used for the training of MSA-based protein language models. It differs from using a decoder, which, though designed to perform autoregressive generation of amino acids to form a new sequence, requires training a full encoder–decoder model and learning a parametric function mapping an MSA to a distribution over its sequences (*Hawkins-Hooker et al., 2021b*). We instead directly employed the representation of protein families captured by the self-supervised model MSA Transformer to generate sequences. More sophisticated sampling methods could be considered along this line (*Goyal et al., 2021*), but our minimal approach already gives very promising results.

We have focused on a large protein language model and compared it to the simplest model capturing coevolution, namely the Potts model, but we note that interpretable models of intermediate complexity such as restricted Boltzmann machines (*Tubiana et al., 2019*) could also be explored for coevolution-driven protein design. All these methods rely on MSAs; this is very useful to capture coevolution, but also means that one has to rely on potentially imperfect alignments. Thus, starting from alignment-free methods (*Bileschi et al., 2022*; *Shin et al., 2021*; *Madani et al., 2021*) also constitutes a promising direction.

## Methods

### Using MSA Transformer to generate sequences via an iterative masking procedure

#### Iterative masking procedure

In order to generate new sequences using MSA Transformer, we directly leverage the model's ability to assign, to arbitrary masked residue positions, a probability for each of the possible amino-acid tokens,

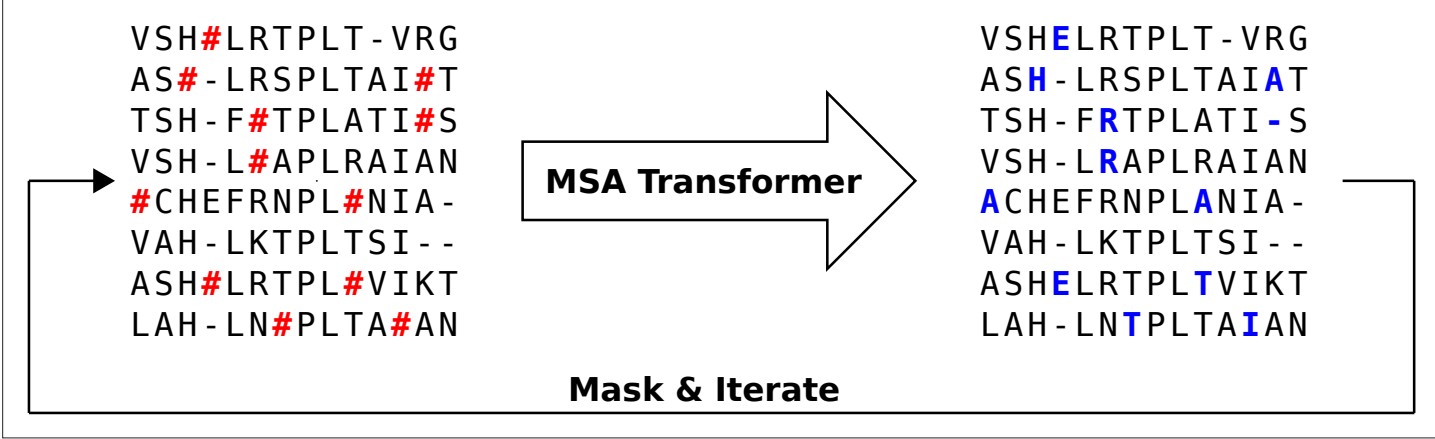

**Figure 7.** Iterative masking procedure to generate sequences using MSA Transformer. Here, the red hashtag (#) stands for a masked amino acid, while blue uppercase letters stand for predicted amino acids at the masked positions.

The online version of this article includes the following figure supplement(s) for figure 7:

**Figure supplement 1.** Evolution of mean scores during the iterative masking procedure, for family PF00153.

**Figure supplement 2.** Evolution of mean scores during the iterative masking procedure, for family PF00096.

**Figure supplement 3.** Evolution of mean scores during the iterative masking procedure, for family PF13354.

**Figure supplement 4.** Evolution of inferred contact maps during the iterative masking procedure, for family PF00153.

given by the softmax of the model's output logits (*Wang and Cho, 2019*; *Goyal et al., 2021*; *Rao et al., 2021b*). Indeed, in its pre-training, MSA Transformer applies the MLM objective to a training set of 26 million MSAs (*Rao et al., 2021b*). For this, it minimizes a pseudolikelihood loss, which reads, for an MSA $\mathcal{M}$, and a version $\widetilde{\mathcal{M}}$ of $\mathcal{M}$ in which some amino acids (those in a 'mask') are masked:

$$\mathcal{L}_{\mathrm{MLM}}(\mathcal{M}, \widetilde{\mathcal{M}}; \theta) = - \sum_{(m,i) \in \mathrm{mask}} \log p(x_{m,i} \mid \widetilde{\mathcal{M}}; \theta).$$

(1)

Here, $x_{m,i}$ denotes the amino acid at the $i$th residue position in the $m$th sequence of $\mathcal{M}$, and $\theta$ denotes all model parameters. For each position $i$ in each sequence $m$, the model outputs one value ('logit') per amino-acid/gap symbol, and softmax-normalizing all values from this location in the MSA yields the conditional probabilities $p(x_{m,i}|\widetilde{\mathcal{M}}; \theta)$ in *Equation 1*, which are then summed over the subset of masked MSA locations.

We propose an iterative masking procedure (see *Figure 7*) which, given an arbitrary MSA $\mathcal{M}$ of natural sequences, proceeds as follows:

1. If necessary, subsample $\mathcal{M}$ to obtain an input MSA $\mathcal{M}'$ for MSA Transformer. The depth of $\mathcal{M}'$ is chosen given the memory footprint of MSA Transformer. In practice, we use input MSAs containing 600 sequences picked uniformly at random from our natural MSA. (During training, the authors of *Rao et al., 2021a* kept $LM < 2^{14}$, where $L$ is sequence length and $M$ is MSA depth. However, we found that during inference we can use $2^{17}$ tokens on an Nvidia V100 32GB GPU.) Note that, for large protein families, multiple 600-sequence MSAs obtained using the procedure presented here are then combined into a single MSA of the same depth as the natural one (see below).
2. Randomly mask each residue of $\mathcal{M}'$ with a masking probability $p$, otherwise leave it unchanged. In practice, we choose $p = 0.1$ (see 'Choosing parameters in the iterative masking procedure').
3. Feed the masked MSA to the model, and fill each masked entry with the token with highest probability (obtained from the model's output logits).
4. Repeat Steps 2–3 a number of times. In practice, we stop the algorithm after $I = 200$ iterations.

As natural MSAs, we use Pfam full MSAs for 14 protein families, described in 'Datasets'. For each natural MSA $\mathcal{M}$, we repeat the procedure above multiple times, sampling sequences each time from $\mathcal{M}$ without replacement to obtain a different input MSA $\mathcal{M}'$ in Step 1, until all the sequences in $\mathcal{M}$ are used. Note that sequences remain aligned at all times during the procedure. Combining the MSAs resulting from all these batches then yields a synthetic MSA with the same depth as the natural one,

which ensures that the statistical properties of the synthetic MSA are subject to the same magnitude of finite-size errors as those of the natural MSA.

## Choosing parameters in the iterative masking procedure

*Figure 7—figure supplement 1* illustrates, in the case of Pfam family PF00153, for different values of the masking probability $p$, how different properties of the generated MSAs evolve with the number $I$ of iterations in the iterative masking procedure. For $p < 0.5$, we observe a gradual divergence from the initial natural sequences (*Figure 7—figure supplement 1A, B*) and a simultaneous increase of scores (*Figure 7—figure supplement 1C, D*, see 'Scoring individual sequences' for definitions) and decrease of MSA diversity (*Figure 7—figure supplement 1E*), and then a saturation of these various measures, as $I$ increases. Our choice $I = 200$ is motivated by the fact that plateaus are reached at this point. However, the final values of all scores depend on $p$. *Figure 7—figure supplement 4* shows the contact maps inferred by MSA Transformer (using the logistic regression on tied row attentions trained in *Rao et al., 2021a*) from generated sequences, for various values of $I$ and $p$, in the case of family PF00153. We observe that the contact map characteristic of the protein family of interest gets gradually lost as $I$ is increased for larger values of $p$ (see *Figure 7—figure supplement 1F* and *Figure 7—figure supplement 4*). These issues when $p$ is increased are understandable, given that the pseudolikelihood loss used for the MLM objective in MSA Transformer ignores dependencies between masked entries. We note that despite this, larger values of p yield overall better average HMMER scores (*Eddy, 1998*) and statistical energy scores (for $p < 0.5$). Our choice of $p = 0.1$ is motivated by the fact that this value is close to that employed in the training of the model ($p \approx 0.12$) (*Rao et al., 2021a*), and that it better preserves contact maps. The product $pI$ gives the average number of times that each amino acid of the MSA is changed during the generation process. With our choices, each amino acid is masked 20 times on average.

The behaviors observed in *Figure 7—figure supplement 1* for PF00153 are generic across the protein families we studied, as can be seen in *Figure 7—figure supplement 2* and *Figure 7—figure supplement 3*, which show the same data as in *Figure 7—figure supplement 1* for Pfam families PF00096 and PF13354 (which have different sequence lengths). This demonstrates that our sequence generation method is robust. In particular, as the parameters $p = 0.1$ and $I = 200$ yield satisfactory convergence of MSA properties and preservation of contact maps in all cases, we used these parameters throughout, without any family-specific fine-tuning.

The sequences thus generated by our method do not coincide with natural ones. The fraction of MSA-Transformer–generated sequences which are identical to sequences in the input natural MSAs is below $5 \times 10^{-4}$ for all large families considered, except three families with low diversity and/or very short sequence length (PF00096, PF00397, and PF00595).

## Variants of the iterative masking procedure

In our algorithm, we mask tokens randomly throughout the input MSA. We also explored an alternative procedure where masking is restricted to the first sequence of the input MSA. Thus, all other sequences act as a context for the first sequence which is gradually modified. This can be done either with a fixed context, or by sampling different sequences from the natural MSA at each iteration to form a variable context. Note that the procedure with fixed context is reminiscent of the non-iterative one used in *Meier et al., 2021* to compute DMS scores from MSA Transformer. For the same masking probability p = 0.1 as in our standard procedure (note that fewer iterations are needed for convergence, in practice $I = 20$ suffices), the alternative procedure with fixed context yields sequences that are overall slightly less different from natural ones than the standard iterative masking procedure, while the opposite holds with variable context. Besides, both alternative procedures yield sequences with better HMMER scores, but worse statistical energy scores, than natural ones – see *Appendix 1—table 3*. Finally, the two- and three-body statistics (defined in 'Analyzing the statistics of MSAs') of the natural MSA are less well reproduced using these alternative procedures than the standard one – see *Appendix 1—table 3*. We also note that these variants are computationally more demanding. In this context, we decided to focus on the standard iterative masking procedure.

There are also different ways of selecting the token to fill each masked position. We have chosen a greedy sampling method where the token with highest probability is selected. We also explored an alternative method where the new token to fill the masked position is chosen by sampling the

probability distribution given by the softmax of the logits, see **Equation 1**. This method allows to introduce a sampling temperature $T$ into the softmax operation and compute the probability as $\mathrm{p} = \mathrm{softmax}(\boldsymbol{\xi}/\mathrm{T})$, where $\boldsymbol{\xi}$ is the logit vector. Note that the greedy method that we employ corresponds to sampling at $T = 0$. We found that MSAs generated with higher values of $T$ are farther from the corresponding natural MSAs, showing that increasing this sampling temperature promotes originality. However, they are of lower quality according to our HMMER and statistical energy scores, and reproduce the statistics of the natural data less well. These results, summarized in **Appendix 1—table 3**, motivated us to mainly consider greedy sampling.

Finally, in our iterative masking procedure, we subsample the initial natural MSAs uniformly at random. We also tried diversity maximizing sampling (**Rao et al., 2021a**), but we found that random sampling gives slightly better results.

## Sampling sequences from Potts models

To sample independent equilibrium sequences from Potts models, we used the strategy described in **Lupo et al., 2022**. Specifically, we fitted Potts models on each of our natural MSAs using bmDCA (**Figliuzzi et al., 2018**) (https://github.com/ranganathanlab/bmDCA; **Figliuzzi and Barrat-Charlaix, 2020**). Using bmDCA is known to yield Potts models with good generative power (**Figliuzzi et al., 2018**; **Russ et al., 2020**).

Consider a sequence of $L$ amino-acid sites. We denote by $x_i \in \{1, \ldots, q\}$ the state of site $i \in \{1, \ldots, L\}$, where $q = 21$ is the number of possible states, namely the 20 natural amino acids and the alignment gap. The Potts model Hamiltonian of a sequence $\boldsymbol{x} = (x_1, \ldots, x_L)$ reads (**Weigt et al., 2009**; **Cocco et al., 2018**):

$$H(\boldsymbol{x}) = -\sum_{i=1}^{L} h_i(x_i) - \sum_{j=1}^{L} \sum_{i=1}^{j-1} e_{ij}(x_i, x_j). \tag{2}$$

For each MSA $\mathcal{M}$ in **Appendix 1—table 5**, we inferred parameters $h_i(x_i)$ and $e_{ij}(x_i, x_j)$ by bmDCA (**Figliuzzi et al., 2018**; **Russ et al., 2020**). The Potts model probability distribution is then given by the Boltzmann distribution associated to the Hamiltonian $H$ in **Equation 2**:

$$P(\boldsymbol{x}) = \frac{e^{-H(\boldsymbol{x})/T}}{Z}, \tag{3}$$

where $Z$ is a constant ensuring normalization and $T$ is a parameter whose default value is 1. To generate a synthetic MSA from $\mathcal{M}$, we performed equilibrium MCMC sampling from the Potts model with Hamiltonian $H$ in **Equation 2**. Specifically, we used the implementation in **Lupo et al., 2022** of the Metropolis–Hastings algorithm, in which each step is a proposed mutation at a single amino-acid site. We started from a set of $M$ randomly and independently initialized sequences, where $M$ is the depth of $\mathcal{M}$, and made a total number $N$ of Monte Carlo steps on each sequence. For each $\mathcal{M}$, suitable values for $N$ are estimated by bmDCA during its training, to ensure that Metropolis–Hastings sampling reaches thermal equilibrium after $N$ steps when starting from a randomly initialized sequence (**Figliuzzi et al., 2018**). We thus used the value of $N$ estimated by bmDCA at the end of training. This yielded, for each MSA in **Appendix 1—table 5**, a synthetic MSA of the same depth, composed of independent equilibrium sequences.

This procedure allows to tune the sampling temperature $T$, in a similar spirit as for MSA Transformer, cf. 'Variants of the iterative masking procedure'. This amounts to tuning the selection strength. Recall that Potts models are inferred at $T = 1$, which is thus the default value. Using MCMC sampling as described above, we first generated synthetic MSAs at $T = 1$, and using regularization strength $\lambda = 10^{-2}$. These correspond to the default parameters $(\lambda, T)$, matching those employed in **Figliuzzi et al., 2018**, and allowing direct comparison with those results. Importantly, using sampling temperature $T = 1$ means that the distribution learnt from natural data is directly sampled. However, it was found in **Russ et al., 2020** that sequences generated at $T = 1$ have worse statistical energy scores than natural sequences, due at least in part to high regularization, and that this can be corrected by lower-temperature sampling. Therefore, for completeness, we also considered all parameter combinations $(\lambda, T)$ used in **Russ et al., 2020** for PF00072. **Appendix 1—table 4** shows that decreasing sampling temperature strongly improves the mean statistical energy score, as it should, and somewhat improves

HMMER scores and structural scores. However, this comes at the cost of decreasing MSA diversity and getting sequences substantially more similar to natural ones. It also strongly impairs the fitting of the one- and two-body statistics. The effect of changing regularization strength (at inference) appears to be more minor, but decreasing it allows to somewhat mitigate the loss of diversity associated to lowering temperature. In light of these results, and to make our comparison to bmDCA comprehensive, we used $(\lambda, T) = (10^{-3}, 0.33)$ (*Russ et al., 2020*) in addition to $(\lambda, T) = (10^{-2}, 1)$ (*Figliuzzi et al., 2018*) throughout our analysis of deep MSAs. In the case of shallow MSAs (3), we employed $(\lambda, T) = (10^{-2}, 0.33)$ instead of $(\lambda, T) = (10^{-3}, 0.33)$ because shallow MSAs require stronger regularization strengths.

## Scoring individual sequences

We use different scores to compare natural and generated sequences.

First, HMMER scores (*Eddy, 1998*) are computed, for each sequence, from the Pfam profile Hidden Markov Models (HMM), employing the function `hmmsearch` from the HMMER Suite version 3.3.2 (http://hmmer.org). HMMER scores are homology scores, which are in particular used in Pfam to search sequence databases for sequence homologs and to construct full MSAs starting from curated seed MSAs. Higher HMMER scores indicate better matches to the Pfam HMM.

Second, DCA statistical energy scores are computed for each sequence using the Potts model Hamiltonian $H$ in *Equation 2* with the couplings and the fields inferred by bmDCA on the natural MSA of the family of interest (see 'Sampling sequences from Potts models'). The statistical energy score is then defined as the opposite of the statistical energy, that is $-H(x)$ for a sequence $x$, so that, here too, higher values mean better scores.

We also compute AlphaFold (*Jumper et al., 2021*) structural prediction confidence scores, that is pLDDT values. Given the computational cost, for each natural or generated MSA, we evaluate pLDDT values for a subset of 200 randomly sampled sequences.

Finally, we compute the root-mean-square deviation (RMSD) between a reference experimental structure of the family of focus (see list in *Appendix 1—table 5*) and the AlphaFold predicted structures, also for a subset of 200 randomly sampled sequences in each MSA.

Because AlphaFold takes MSAs as input, we compute these two structural scores using the whole natural MSA of the family of interest as context in all cases. In addition, for the protein family PF00072, we also used fully synthetic MSAs as input to AlphaFold. Structural scores are then very similar to those obtained using natural context (see *Appendix 1—table 4*).

## Analyzing the statistics of MSAs

To compare the generated MSAs to the natural ones, we consider different statistical measures.

First, to analyze how faithfully the generated MSAs reproduce the statistics of the natural ones at various orders, we compute the r20 score (*Haldane et al., 2018*; *McGee et al., 2021*). Specifically, to obtain *Figure 4*, we analyze the frequency of subsequences spanning 2–10 non-contiguous columns. In each of 1000 randomly sampled sets of columns for each subsequence length, we compute the frequency of the 20 most frequent words in natural and synthetic MSAs of the family considered, and evaluate the Pearson correlation between these top 20 frequencies in the MSA of focus and those in a reference MSA. We then average these Pearson correlation values over all sets of 1000 columns, yielding the r20 score.

To further inspect low-order statistics, in each MSA, we compute the one-body frequencies of occurrence of each amino acid at each site, the two-body frequencies of each pair of amino acids at each pair of sites, and the three-body frequencies associated to triplets. We denote them by $f_i(x)$, $f_{ij}(x, y)$, $f_{ijk}(x, y, z)$, where $i$, $j$, and $k$ denote sites, while $x$, $y$, and $z$ represent amino acids (see 'Sampling sequences from Potts models'). We then estimate the second- and third-order connected correlations as:

$$C_{ij}(x, y) = f_{ij}(x, y) - f_i(x)f_j(y); \tag{4}$$

$$C_{ijk}(x, y, z) = f_{ijk}(x, y, z) - f_{ij}(x, y)f_k(z) - f_{ik}(x, z)f_j(y) - f_{jk}(y, z)f_i(x) + 2f_i(x)f_j(y)f_k(z). \tag{5}$$

We also compute the 'plug-in' estimates of the Shannon entropy of each site $H_i$, and of the two- and three-body joint entropies $H_{ij}$ and $H_{ijk}$, from the frequencies. They yield the plug-in estimates of the mutual information $I_{ij}$ between two columns, and of the *co-information* $I_{ijk}$ between three columns:

$$I_{ij} = H_i + H_j - H_{ij}\,; \tag{6}$$

$$I_{ijk} = H_i + H_j + H_k - H_{ij} - H_{ik} - H_{jk} + H_{ijk}\,. \tag{7}$$

Co-information is a measure of higher-order statistical dependencies (*McGill, 1954*; *Timme et al., 2014*; *Quax et al., 2017*; *Rosas et al., 2019*), which generalizes mutual information to triplets of random variables, vanishes for independent variables, and reflects the balance between redundancy and synergy in these triplets (*Williams and Beer, 2010*; *Rosas et al., 2016*). A systematic finite-size error occurs when estimating entropies using the plug-in estimate from frequencies measured in finite datasets (*Bialek, 2012*), and it affects entropy-derived quantities such as mutual information and co-information. Here, we do not attempt to correct it. Rather, we only make comparisons between MSAs of the same length and depth, which are affected by the same finite-size errors.

## Characterizing the distribution of sequences in MSAs

Another way of studying the properties of generated MSAs is to analyze the distribution of their sequences in sequence space, and to compare it to that of natural sequences in the same family.

First, to assess whether generated sequences most resemble natural ones that are well represented in their family or, rather, rare ones, we consider for each synthetic sequence its closest natural sequence. We then count the number of neighbors of this natural sequence in the natural MSA, that is the number of natural sequences that have (normalized) Hamming distance below $\delta = 0.2$ with the sequence of interest. Note that the inverse of this number of neighbors gives the sequence weight $w_i$ introduced in *Equation 8*.

Second, to explore the distributions in sequence space of sequences within each MSA, and compare synthetic and natural MSAs, we associate to each sequence the concatenation of the one-hot encodings of each of its amino acids (*Figliuzzi et al., 2018*). We perform a PC analysis of the matrix corresponding to the natural MSA in this representation. We can then represent natural and synthetic sequences as points projected in the space defined by the first two PCs of the natural MSA.

Third, to analyze in more detail the apparent relatedness of generated sequences, and compare it to real phylogenetic relationships in natural sequences, we infer phylogenetic trees from each MSA using `Fast-Tree 2` (*Price et al., 2010*). To quantitatively compare the topologies of these trees, which do not have the same leaves, we analyze the eigenvalue spectrum of their MGL (*Lewitus and Morlon, 2016*). The MGL of a phylogenetic tree is defined as the difference between its degree matrix (a diagonal matrix whose $i$th diagonal entry is the sum of the branch lengths from node $i$ to all other nodes in the tree) and the matrix of patristic distances (whose $(i, j)$ th entry is the branch length between nodes $i$ and $j$). Given the computational cost of running such an analysis on our deep MSAs, we use a bootstrap-aggregating strategy in the spirit of *Colijn and Plazzotta, 2018*. Namely, for each MSA we compute 200 different trees, each one inferred from a different sub-MSA of 500 sequences, itself randomly sampled from the whole MSA. Then, for each of these trees, we compute the eigenvalue spectrum of the MGL. Next, we merge all these spectra together to obtain a single eigenvalue spectral density. Note that this method has the advantage of not depending on the details of the topology of one large inferred tree, which are known to be sensitive to the choice of phylogeny reconstruction algorithm.

## Datasets

To generate synthetic MSAs with MSA Transformer and bmDCA and compare them to their natural counterparts, we consider the deep Pfam 'full' alignments (*Mistry et al., 2021*) associated to 14 different protein domains (*Appendix 1—table 5*). Each MSA is a matrix $\mathcal{M}$ with $L$ columns, representing the different amino-acid sites, and $M$ rows. Each row $i$, denoted by $\boldsymbol{x}^{(i)}$, represents one sequence of the alignment. We refer to $L$ as the MSA length, and to $M$ as its depth. For all our MSAs, $M > 36000$. These alignments are the same as in *Lupo et al., 2022*, except that we removed PF13354 (Beta-lactamase2) from this set of deep MSAs because of its smaller depth. However, this family is included in our additional analyses (see *Appendix 1—table 6*).

Deep MSAs generally include some highly similar sequences due to phylogenetic relatedness. This can be characterized via the effective depth (*Weigt et al., 2009*)

$$M_{\mathrm{eff}}^{(\delta)} := \sum_{i=1}^{M} w_i, \quad \text{with} \quad w_i := \left| \left\{ i' : d_{\mathrm{H}}(\boldsymbol{x}^{(i)}, \boldsymbol{x}^{(i')}) < \delta \right\} \right|^{-1}, \tag{8}$$

where $d_H(x, y)$ is the (normalized) Hamming distance between two sequences $x$ and $y$, that is the fraction of sites where the amino acids differ, and we set $\delta = 0.2$. Note that the inverse of the sequence weight $w_i$ in *Equation 8* is the number of neighbors in 'Characterizing the distribution of sequences in MSAs', and that $M_{\text{eff}}^{(0.2)}/M$ can be as low as 0.06 for our natural MSAs.

All these families were previously shown to be well fitted by Potts models inferred by bmDCA (*Figliuzzi et al., 2018*), making our results on sequence generation by bmDCA readily comparable with previous results. Our domains' short lengths are convenient because bmDCA is computationally demanding, and also in view of MSA Transformer's large memory footprint, which is $O(LM^2) + O(L^2)$. Furthermore, their large depth is crucial to our comparisons, as it allows Potts models to be accurately fitted (*Figliuzzi et al., 2018*).

We extended our study to small protein families by considering seven additional families, listed in *Appendix 1—table 6*, for which we also started from Pfam 'full' MSAs. These families comprise from nine to a few hundreds of sequences. We also considered two additional protein families, also listed in *Appendix 1—table 6*, for our comparison with published experimental datasets.

## Acknowledgements

This project has received funding from the European Research Council (ERC) under the European Union's Horizon 2020 research and innovation programme (grant agreement no. 851173, to A-FB).

## Additional information

### Funding

| Funder | Grant reference number | Author |
| --- | --- | --- |
| European Research Council | 851173 | Damiano Sgarbossa<br>Umberto Lupo<br>Anne-Florence Bitbol |

The funders had no role in study design, data collection, and interpretation, or the decision to submit the work for publication.

### Author contributions

Damiano Sgarbossa, Conceptualization, Software, Validation, Investigation, Visualization, Methodology, Writing – original draft; Umberto Lupo, Conceptualization, Resources, Supervision, Methodology, Writing – review and editing; Anne-Florence Bitbol, Conceptualization, Supervision, Funding acquisition, Methodology, Writing – original draft, Writing – review and editing

### Author ORCIDs

Damiano Sgarbossa http://orcid.org/0000-0002-7878-6061
Umberto Lupo http://orcid.org/0000-0001-6767-493X
Anne-Florence Bitbol http://orcid.org/0000-0003-1020-494X

### Decision letter and Author response

Decision letter https://doi.org/10.7554/eLife.79854.sa1
Author response https://doi.org/10.7554/eLife.79854.sa2

## Additional files

### Supplementary files
• MDAR checklist

### Data availability

Python code for generating sequences using the iterative masking procedure is archived at: https://doi.org/10.5281/zenodo.7684052. Raw data were collected from two public sources: (1) MSAs from the Pfam database (https://pfam.xfam.org/); (2) further MSAs from https://github.com/matteofigliuzzi/bmDCA (*Barrat-Charlaix, 2017*). We generated sequences with bmDCA using code publicly available at https://github.com/ranganathanlab/bmDCA (*Figliuzzi and Barrat-Charlaix, 2020*).

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

# Appendix 1

**Appendix 1—table 1.** p values of the Kolmogorov–Smirnov test comparing the distributions of homology, coevolution, and structure-based scores across natural and synthetic multiple sequence alignments (MSAs).

For each score except the root-mean-squared deviation (RMSD), we test the null hypothesis that the scores of MSA-Transformer–generated sequences are greater or equal than those of Boltzmann machine DCA (bmDCA)-generated sequences, in the (stringent) sense that the cumulative distribution function of the former is always below that of the latter. Here, bmDCA1 stands for bmDCA with $(\lambda, T) = (10^{-3}, 0.33)$ and bmDCA2 for bmDCA with $(\lambda, T) = (10^{-2}, 1)$. For the RMSD, the null hypothesis is that the scores of MSA-Transformer–generated sequences are smaller or equal than those of bmDCA-generated sequences (recall that smaller RMSDs are better). In all cases, a p value close to one (resp. zero) means that the null hypothesis tested should be accepted (resp. rejected). Reported zero p values are too small to be properly assessed by the algorithm.

| | HMMER score | | Statistical energy score | | pLDDT confidence | | RMSD | |
|---|---|---|---|---|---|---|---|---|
| **Pfam ID** | **MSA Tr. ≥ bmDCA1** | **MSA Tr. ≥ bmDCA2** | **MSA Tr. ≥ bmDCA1** | **MSA Tr. ≥ bmDCA2** | **MSA Tr. ≥ bmDCA1** | **MSA Tr. ≥ bmDCA2** | **MSA Tr. ≤ bmDCA1** | **MSA Tr. ≤ bmDCA2** |
| PF00004 | 0 | 1.0 | 0 | 1.0 | 0.20 | 1.0 | 0.33 | 0.96 |
| PF00005 | 0.93 | 1.0 | 0 | 1.0 | $6.5 \cdot 10^{-19}$ | 1.0 | 0.78 | 0.96 |
| PF00041 | 0.99 | 1.0 | 0 | 1.0 | 1.0 | 1.0 | $2.9 \cdot 10^{-14}$ | $5.9 \cdot 10^{-3}$ |
| PF00072 | $4.7 \cdot 10^{-12}$ | 1.0 | 0 | 1.0 | $9.1 \cdot 10^{-6}$ | 1.0 | $3.4 \cdot 10^{-6}$ | 0.96 |
| PF00076 | $9.5 \cdot 10^{-134}$ | 1.0 | 0 | 1.0 | $6.5 \cdot 10^{-19}$ | 1.0 | $9.2 \cdot 10^{-5}$ | 1.0 |
| PF00096 | 0.04 | 1.0 | 0 | 1.0 | 0.01 | 0.92 | 0.92 | $2.4 \cdot 10^{-5}$ |
| PF00153 | 0.91 | 1.0 | 0 | 1.0 | 0.98 | 0.84 | $9.3 \cdot 10^{-10}$ | 1.0 |
| PF00271 | $5.9 \cdot 10^{-30}$ | 1.0 | 0 | 1.0 | 0.33 | 1.0 | 0.02 | 0.88 |
| PF00397 | 0 | 1.0 | 0 | 1.0 | $1.5 \cdot 10^{-3}$ | 1.0 | $4.8 \cdot 10^{-10}$ | 0.38 |
| PF00512 | 0 | 1.0 | 0 | 1.0 | $4.3 \cdot 10^{-3}$ | 1.0 | 0.96 | 0.67 |
| PF00595 | 0.83 | 1.0 | 0 | 1.0 | $1.0 \cdot 10^{-15}$ | 1.0 | $7.2 \cdot 10^{-24}$ | 1.0 |
| PF01535 | 0.98 | 1.0 | 0 | 1.0 | 1.0 | 1.0 | 0.78 | $4.1 \cdot 10^{-8}$ |
| PF02518 | 0 | 1.0 | 0 | 1.0 | 1.0 | 1.0 | 1.0 | 1.0 |
| PF07679 | 1.0 | 1.0 | 0 | 1.0 | 1.0 | 1.0 | $4.3 \cdot 10^{-3}$ | 1.0 |

**Appendix 1—table 2.** Median homology, coevolution, and structure-based scores in natural and synthetic sequences.

We report the median values of each of the scores shown in *Figure 1*, as well as their standard deviations (between parentheses), for natural sequences ('Nat.'), for sequences generated by our method based on MSA Transformer, and for sequences generated by Boltzmann machine DCA (bmDCA) at low temperature, that is with $(\lambda, T) = (10^{-3}, 0.33)$ (denoted by 'bmDCA'). Scores are normalized as *Figure 1*, except that, for statistical energy, we subtract the median of natural scores instead of the mean for clarity (therefore, all natural MSAs have median 0 and standard deviation 1 for this score). For all scores, the best median among those of the two synthetic MSAs is shown in bold, and for the predicted local-distance difference test (pLDDT) score, it is shown in red if it is better than that the other synthetic MSA by a margin larger than the largest standard deviation. Recall that higher values are better for all scores, except root-mean-squared deviation (RMSD), for which the opposite holds.

| Pfam ID | HMMER score | | | Statistical energy score | | | pLDDT confidence (%) | | | RMSD (Å) | | |
|---|---|---|---|---|---|---|---|---|---|---|---|---|
| | Nat. | MSA Tr. | bmDCA | Nat. | MSA Tr. | bmDCA | Nat. | MSA Tr. | bmDCA | Nat. | MSA Tr. | bmDCA |
| PF00004 | 0.5 (0.2) | 0.6 (0.2) | **0.8** (0.2) | 0 (1) | 0.8 (0.9) | **1.6** (0.1) | 85.4 (4.1) | **85.8** (4.5) | 81.7 (0.7) | 3.4 (0.8) | **2.8** (0.7) | 3.6 (0.5) |
| PF00005 | 0.7 (0.1) | 0.8 (0.1) | 0.8 (0.1) | 0 (1) | 1.8 (0.9) | **3.1** (0.2) | 83.0 (6.9) | 89.0 (4.2) | **91.6** (1.6) | 3.8 (1.2) | 2.8 (1.0) | 2.8 (0.8) |
| PF00041 | 0.6 (0.1) | **0.9** (0.2) | 0.5 (0.1) | 0 (1) | 1.5 (1.0) | **4.9** (0.5) | 90.0 (4.5) | 92.0 (3.2) | 79.2 (2.7) | 2.1 (2.1) | **2.9** (0.5) | 3.4 (2.2) |
| PF00072 | 0.7 (0.1) | **0.9** (0.1) | 0.8 (0.1) | 0 (1) | 2.1 (0.8) | **3.8** (0.3) | 94.5 (3.4) | **94.9** (1.9) | 94.1 (0.5) | 2.4 (0.3) | 2.3 (0.1) | **2.1** (0.1) |
| PF00076 | 0.6 (0.1) | 0.8 (0.2) | 0.8 (0.1) | 0 (1) | 1.5 (0.8) | **3.5** (0.2) | 82.2 (4.4) | 84.6 (4.8) | **87.6** (1.5) | 1.8 (0.5) | 1.4 (0.6) | 1.4 (0.1) |
| PF00096 | 0.8 (0.0) | **0.9** (0.0) | 0.9 (0.0) | 0 (1) | 2.2 (0.8) | **2.8** (0.3) | 93.0 (2.0) | **94.0** (0.8) | 93.7 (0.2) | 0.6 (0.1) | 0.4 (0.1) | 0.4 (0.0) |
| PF00153 | 0.5 (0.1) | **0.6** (0.1) | 0.5 (0.1) | 0 (1) | 0.6 (0.8) | **2.6** (0.3) | 65.0 (5.0) | **66.6** (6.2) | 64.9 (4.3) | 5.1 (1.8) | 4.4 (1.5) | 4.3 (1.1) |
| PF00271 | 0.5 (0.1) | 0.5 (0.1) | 0.5 (0.2) | 0 (1) | 1.0 (0.9) | **2.4** (0.3) | 78.4 (4.6) | **86.4** (5.4) | 83.8 (2.2) | 2.0 (0.8) | 2.3 (0.6) | **1.8** (0.1) |
| PF00397 | 0.7 (0.1) | **0.8** (0.1) | 0.8 (0.1) | 0 (1) | 0.5 (0.9) | **2.1** (0.2) | 88.1 (2.2) | **88.9** (2.4) | 88.2 (1.0) | 0.9 (0.3) | 0.9 (0.3) | 0.9 (0.1) |
| PF00512 | 0.5 (0.1) | 0.8 (0.2) | 0.7 (0.2) | 0 (1) | 1.5 (1.0) | **3.2** (0.3) | 91.0 (4.0) | **90.2** (4.0) | 89.5 (1.5) | 2.1 (0.6) | **2.2** (0.5) | 3.1 (0.2) |
| PF00595 | 0.7 (0.1) | 0.7 (0.1) | 0.7 (0.1) | 0 (1) | 0.5 (0.9) | **2.6** (0.3) | 93.4 (4.5) | 94.0 (1.8) | **95.1** (0.8) | 1.8 (0.4) | 1.7 (0.5) | **1.4** (0.2) |
| PF01535 | 0.5 (0.1) | **0.9** (0.2) | 0.6 (0.1) | 0 (1) | 2.3 (1.1) | **4.1** (0.2) | 82.4 (6.2) | 94.3 (5.5) | 77.9 (3.6) | 1.0 (1.1) | **0.4** (0.7) | 0.5 (0.4) |
| PF02518 | 0.6 (0.2) | **0.8** (0.2) | 0.7 (0.2) | 0 (1) | 1.9 (0.9) | **3.5** (0.2) | 88.0 (6.0) | 91.0 (6.3) | 73.6 (2.3) | 4.1 (0.9) | **3.9** (0.5) | 4.7 (1.1) |
| PF07679 | 0.5 (0.1) | **0.7** (0.2) | 0.4 (0.1) | 0 (1) | 1.7 (1.0) | **5.2** (0.6) | 93.5 (3.8) | 95.3 (2.9) | 89.8 (2.2) | 1.3 (1.0) | 1.2 (0.5) | 1.2 (0.2) |

**Appendix 1—table 3.** Comparing different generation methods of MSA Transformer. Various scores are shown for the natural MSA of protein family PF00153 and for synthetic MSAs generated in different ways from this family (each synthetic MSA comprises 10,000 sequences). For generation using MSA Transformer (see 'Using MSA Transformer to generate sequences via an iterative masking procedure'), our standard iterative masking procedure is shown with its default greedy sampling (corresponding to $T = 0$) and two higher temperatures. Variants of the procedure where only the first sequence is masked ('Context', either fixed or variable, both with greedy sampling) are also shown. We report the mean Hamming distance to the closest natural sequence, which is not itself in the case of natural sequences ('Distance') as well as the mean HMMER and statistical energy scores ('-Energy') described in 'Scoring individual sequences'. Note that statistical energy scores are shifted by the mean value obtained for the natural MSA (which is −235.8). We also report the Pearson correlations between the two- and three-body statistics of the natural and the generated MSAs, denoted, respectively, by $\rho\left[C_{ij}\right]$ and $\rho\left[C_{ijk}\right]$ (for the natural MSA we report the Pearson correlation between two halves of this MSA), as illustrated in *Figure 4—figure supplement 2*.

| | | MSA Transformer | | | | |
| | | Iterative masking | | | Context (greedy) | |
| Score | Natural sequences | Greedy | $T = 0.5$ | $T = 1.0$ | Fixed | Variable |
|---|---|---|---|---|---|---|
| Distance | 0.155 | 0.271 | 0.305 | 0.514 | 0.232 | 0.262 |
| HMMER | 48.0 | 58.2 | 58.1 | 48.4 | 58.7 | 63.8 |
| − Energy | 0 | 13.0 | 8.5 | −42.0 | −15.4 | −13.2 |
| $\rho\left[C_{ij}\right]$ | 0.94 | 0.84 | 0.84 | 0.62 | 0.73 | 0.81 |
| $\rho\left[C_{ijk}\right]$ | 0.89 | 0.80 | 0.76 | 0.41 | 0.66 | 0.77 |

**Appendix 1—table 4.** Impact of regularization strength and sampling temperature on sequence generation by Boltzmann machine DCA (bmDCA), for family PF00072.

We compare MSAs obtained using bmDCA with different regularization strengths $\lambda$ (for inference) and sampling temperatures $T$ (for generation) with the natural and the MSA-Transformer–generated MSAs. In each case, we report the average of the Hamming distances of each sequence to their closest natural neighbor, which is not itself in the case of natural sequences ('Distance'), as well as the effective MSA depth, the scores defined in 'Scoring individual sequences', and the Pearson correlation coefficients of the two- and three-body connected correlations computed from natural and generated MSAs ($\rho\left[C_{ij}\right]$ and $\rho\left[C_{ijk}\right]$). For MSA Transformer ('MSA Tr.'), bmDCA (0.01,1) and bmDCA (0.001,0.33), we also computed structural scores by feeding the entire synthetic MSA to Alphafold as context MSA (instead of using the natural MSA as context, see 'Scoring individual sequences'). Structural scores are then very similar to those obtained using natural context.

| Type | $\lambda$ | $T$ | Distance | $M_{\text{eff}}^{(0.2)}$ | HMMER | - Energy | $\rho\left[C_{ij}\right]$ | $\rho\left[C_{ijk}\right]$ | pLDDT (%) | RMSD (Å) |
|---|---|---|---|---|---|---|---|---|---|---|
| Natural | - | - | 0.193 | 40,180 | 90.3 | 0 | 0.99 | 0.88 | 93.6 | 2.5 |
| MSA Tr. | - | - | 0.348 | 9304 | 119.1 | 59.1 | 0.73 | 0.53 | 94.7 | 2.35 |
| | | | | Same as above, with synthetic context: | | | | | 95.1 | 2.37 |
| bmDCA | 0.01 | 1 | 0.55 | 73,062 | 66.5 | −37.0 | 0.96 | 0.58 | 84.3 | 2.58 |
| | | | | Same as above, with synthetic context: | | | | | 83.9 | 2.70 |
| bmDCA | 0.01 | 0.66 | 0.294 | 18,911 | 101.7 | 92.2 | 0.48 | 0.11 | 94.2 | 2.61 |
| bmDCA | 0.01 | 0.33 | 0.251 | 12 | 103.2 | 118.3 | 0.42 | 0.05 | 94.2 | 2.55 |
| bmDCA | 0.001 | 1 | 0.525 | 73,062 | 86.9 | −18.3 | 0.97 | 0.63 | 89.7 | 2.44 |
| bmDCA | 0.001 | 0.66 | 0.296 | 21,294 | 103.9 | 89.3 | 0.48 | 0.19 | 94.3 | 2.6 |
| bmDCA | 0.001 | 0.33 | 0.274 | 14 | 107.7 | 109.6 | 0.4 | 0.13 | 94.0 | 2.14 |
| | | | | Same as above, with synthetic context: | | | | | 94.2 | 2.24 |

**Appendix 1—table 5.** Pfam families and natural MSAs used in our analysis.

$L$ denotes the length of an MSA, $M$ its depth, and $M_{\mathrm{eff}}^{(0.2)}$ its effective depth with distance threshold $\delta = 0.2$, see *Equation 8*. The reference experimental PDB structures used for our root-mean-squared deviation (RMSD) calculations, and their resolutions ('Resol.'), are also reported.

| Pfam ID | Family name | $L$ | $M$ | $M_{\mathrm{eff}}^{(0.2)}$ | PDB ID | Resol. |
|---------|-------------|-----|-----|---------------------------|--------|--------|
| PF00004 | AAA | 132 | 39,277 | 9049 | 4D81 | 2.40 Å |
| PF00005 | ABC_tran | 137 | 68,891 | 43,881 | 1L7V | 3.20 Å |
| PF00041 | fn3 | 85 | 42,721 | 17,782 | 3UP1 | 2.15 Å |
| PF00072 | Response_reg | 112 | 73,063 | 40,180 | 3ILH | 2.59 Å |
| PF00076 | RRM_1 | 69 | 51,964 | 20,273 | 3NNH | 2.75 Å |
| PF00096 | zf-C2H2 | 23 | 38,996 | 12,581 | 4R2A | 1.59 Å |
| PF00153 | Mito_carr | 94 | 93,776 | 17,859 | 1OCK | 2.20 Å |
| PF00271 | Helicase_C | 111 | 66,809 | 25,017 | 3EX7 | 2.30 Å |
| PF00397 | WW | 31 | 39,045 | 3361 | 4REX | 1.60 Å |
| PF00512 | HisKA | 66 | 154,998 | 67,303 | 3DGE | 2.80 Å |
| PF00595 | PDZ | 82 | 71,303 | 4053 | 1BE9 | 1.82 Å |
| PF01535 | PPR | 31 | 109,064 | 37,514 | 4M57 | 2.86 Å |
| PF02518 | HATPase_c | 111 | 80,714 | 59,189 | 3G7E | 2.20 Å |
| PF07679 | I-set | 90 | 36,141 | 14,611 | 1FHG | 2.00 Å |

**Appendix 1—table 6.** Other Pfam families and natural MSAs used in our analysis.

$L$ denotes the length of an MSA and $M$ its depth. The reference experimental PDB structures used for our root-mean-squared deviation (RMSD) calculations, and their resolutions, are also reported.

| Pfam ID | Family name | $L$ | $M$ | PDB ID | Resol. |
|---------|-------------|-----|-----|--------|--------|
| PF01356 | A_amylase_inhib | 68 | 51 | 1OK0 | 0.93 Å |
| PF03440 | APT | 87 | 14 | 6RO0 | 2.13 Å |
| PF04008 | Adenosine_kin | 154 | 342 | 1WVQ | 1.45 Å |
| PF06351 | Allene_ox_cyc | 175 | 378 | 2BRJ | 1.50 Å |
| PF06355 | Aegerolysin | 131 | 322 | 6MYI | 1.15 Å |
| PF16747 | Adhesin_E | 125 | 31 | 6GUT | 1.63 Å |
| PF18648 | ADPRTs_Tse2 | 155 | 9 | 5AKO | 2.40 Å |
| PF13354 | Beta-lactamase2 | 198 | 4642 | 6QW8 | 1.10 Å |
| - | Chorismate mutase *Russ et al., 2020* | 96 | 1130 | 1ECM | 2.20 Å |

