## [Editor Report]

This important study proposes a method to sample novel sequences from a protein language model that could have exciting applications in protein sequence design. The claims are supported by a solid benchmarking of the designed sequences in terms of quality, novelty and diversity.

---

## [Decision Letter]

**Decision letter after peer review:**

Thank you for submitting your article "Generative power of a protein language model trained on multiple sequence alignments" for consideration by *eLife*. Your article has been reviewed by 2 peer reviewers, and the evaluation has been overseen by Lucy Colwell as Reviewing Editor and José Faraldo-Gómez as Senior Editor. All reviewers have opted to remain anonymous.

Essential revisions:

(1) Please address the technical points raised by each referee below, paying particular attention to those around the generalizability of hyperparameter choices, the training of bmDCA, and the comparison between bmDCA and the MSA transformer and the sampling temperatures used, and the evaluation of sample diversity.

(2) Both reviewers note the lack of experimental validation. Adding some level of experimental validation of the proposed method would significantly improve the manuscript. If this is not possible, an alternative option might be to build a retrospective model evaluation using previously published protein design experimental datasets.

*Reviewer #1 (Recommendations for the authors):*

Sampling hyperparameters:

Figures S1/2 convincingly motivate the choice of parameters for one protein family, but it is unclear whether these would generalize well for other proteins. In particular, it is unclear whether the number of iterations should be the same across protein sizes.

bmDCA training:

In figure 3 middle left panel (comparing MSA covariance versus bmDCA-generated covariance) shows a fairly small regression line slope (0.67). This suggests too large a regularization on couplings and/or an insufficient number of iterations. Have you checked that the algorithm successfully converged, or was the maximum number of iterations reached? What about using a smaller regularization strength (two values were used in Russ et al. Science 2020, one small and one large)? I know that parameter tuning is a weakness of bmDCA, but the proposed method also has free parameters and it is not clear whether they are universal for all protein lengths. This could also explain the discrepancy in the topology of the sequence distribution shown in Figure 5 (weak couplings cannot create "ferromagnetic" multimodal distributions).

Evaluation of sequence quality:

I feel that the comparison of Figure 1 is a bit unfair. On the one hand, the proposed MSA-Transformer sampling method generates samples from the T=0 conditional distribution (according to step 3, the bottom of p9), whereas the bmDCA generates samples from the T=1 distribution. It is known that samples from the T=1 distribution have worse statistical energy scores than natural sequences due to the regularization of fields/couplings and that this must be adequately compensated by low-temperature sampling (Russ et al. Science 2020). The authors argue p13 that low-temperature sampling results in distortion of the first and second-order moments of the data, but I think this is not a practical problem for protein design (by definition, an optimal design protocol will discard all suboptimal amino acids for a given position and only retain the most favorable ones).

The comparison of AlphaFold confidence scores between MSAtransformer-generated and natural sequences is interesting but the results are not discussed at all in the main text. Is there a statistically significant difference between the distributions? Moreover, it is not shown here whether the predicted structures are similar to the native ones. Please provide a fold similarity metric such as backbone RMSD values.

Evaluation of sample diversity:

The authors convincingly show that sequences generated by MSAtransformer differ substantially from natural sequences. However, given that the MSA-Transformer samples are essentially obtained by zero-temperature dynamics, it is important to assess how diverse are the generated sequences. This is partially addressed in Figure S5, but consider using simpler metrics. Can the authors provide basic estimates of sample diversity? (e.g., effective number of sequences as a function of the number of samples).

*Reviewer #2 (Recommendations for the authors):*

I have a series of questions that if resolved could help, in my opinion, to create an improved version of this interesting study.

1. The manuscript describes how MSA Transformers could lead to metrics that are "even better" than the natural sequences. I feel that this statement is a bit misleading as the natural sequences are a baseline to compare. There are no better properties than those of the natural sequences because that is what we observe in nature. I would suggest removing those statements as they might produce confusion. On the other hand, if the authors would provide examples of sequences that are "better" in terms of function, stability, or any other quantifiable metric, then I would agree that these statements would make more sense.

2. The authors mention methods that can be trained in an unsupervised way to predict structure based on a single sequence. I wonder if this statement can be clarified, as it is obvious that those methods use several sequences as training. Are these methodologies based on physical principles instead of learning from sequences?

3. It is not clear to me what is the importance of the correlation between HMMER and Hamming distance, the authors should provide more intuition on why this is a relevant metric. Since those correlations are quite low for both models I am not sure if it contributes to the analysis in a meaningful way.

4. Are the contact maps created from the generated MSA of Transformer better than those of bmDCA? My understanding is that the value of p=0.1 optimizes contact map accuracy, but how are those compared against DCA maps?

5. Haldane et al. (https://doi.org/10.1016/j.bpj.2017.10.028) has proposed how pairwise statistics seem to be enough to recapitulate higher order mutational patterns in natural sequences. Could the authors comment on this, and mention if there is a substantial advantage in capturing 3-body statistics in the process of protein design or generative modeling?

6. It would be interesting to see how Alphafold would perform if some of these generated MSAs are used as input. I don't think it is needed to test it with all the families, but it would be an interesting experiment for a single family.

7. The families used in this study appear to have enough statistics to perform well, this is reasonable, however, for the sake of comparison, it would be interesting to see how the MSA Transformer would compare against bmDCA for a family with just a few sequence members. Are the trends the same? Or do we see a change in performance between the two generative methodologies?

---

## [Author Response]

Essential revisions:(1) Please address the technical points raised by each referee below, paying particular attention to those around the generalizability of hyperparameter choices, the training of bmDCA, and the comparison between bmDCA and the MSA transformer and the sampling temperatures used, and the evaluation of sample diversity.

We thank the editor for highlighting the technical points raised by the reviewers. We have treated all of these points thoroughly. Each of them is explained below in our point-by-point response to each reviewer. Here is a brief summary of how we addressed these points.

– Regarding the generalizability of hyperparameter choices, the values we chose yielded satisfactory convergence of MSA properties and preservation of contact maps for all families considered. This holds true in our revised version, where we considered 9 additional protein families (listed in Table 6), with various lengths and depths. In this context, we chose not to engage in family-specific parameter tuning. We now illustrate this robustness better, in two new figures (Figure 7—figure supplements 2 and 3).

– About the training of bmDCA, we did our best to reproduce the published bmDCA results, and to train bmDCA as in the original papers. In our response to reviewer 1 below, we show quantitative comparisons of the fitting performance of two-body correlations between our bmDCA results and published ones.

– About the comparison between bmDCA and MSA Transformer and the sampling temperatures used, we have now performed a comprehensive comparison of our

MSA-Transformer–generated data not only to bmDCA-generated data at sampling temperature T=1, but also at lower sampling temperatures and regularization strengths, following [Russ et al. 2020]. We also now provide an analysis of the statistical significance of the difference between the distributions of all scores in our various generated datasets in Table 1, using the Kolmogorov-Smirnov test, and in Table 2, focusing on the median and standard deviation of scores.

– Regarding the evaluation of sample diversity, we now provide a study of the effective MSA depth in Figure 1—figure supplement 1, and discuss this point thoroughly.

(2) Both reviewers note the lack of experimental validation. Adding some level of experimental validation of the proposed method would significantly improve the manuscript. If this is not possible, an alternative option might be to build a retrospective model evaluation using previously published protein design experimental datasets.

We thank the editor and the reviewers for these very relevant suggestions. While we were not in a position to conduct experiments, we performed a retrospective model evaluation using previously published protein design experimental datasets, and added a new section about this at the end of Results. There, we present a detailed comparison of our generated sequences to those experimentally validated in [Russ et al. 2020], and a calculation of deep mutational scanning scores for two other protein families. The corresponding data is shown in our new Figure 6 and Figure 6—figure supplements 1 and 2.

Reviewer #1 (Recommendations for the authors):Sampling hyperparameters:Figures S1/2 convincingly motivate the choice of parameters for one protein family, but it is unclear whether these would generalize well for other proteins. In particular, it is unclear whether the number of iterations should be the same across protein sizes.

We thank the reviewer for raising this important question. We agree that optimal hyperparameters could potentially depend on protein family characteristics. Throughout our work, we used the following hyperparameter values: masking probability p=0.1 and number of iterations I=200, because they worked well throughout. To illustrate this robustness better, we produced two new figures (Figure 7—figure supplement 2 and Figure 7—figure supplement 3) similar to Figure S1 (now Figure 7—figure supplement 1) for two other protein families, namely PF00096 and PF13354, which are respectively the shortest (L=23) and the longest (L=198) considered in our study. We also added a paragraph in Methods that discusses this point:

"The behaviors observed in Figure 7—figure supplement 1 for PF00153 are generic across the protein families we studied, as can be seen in Figure 7—figure supplement 2 and Figure 7—figure supplement 3, which show the same data as in Figure 7—figure supplement 1 for Pfam families PF00096 and PF13354 (which have different sequence lengths). This demonstrates that our sequence generation method is robust. In particular, as the parameters p = 0.1 and I = 200 yield satisfactory convergence of MSA properties and preservation of contact maps in all cases, we used these parameters throughout, without any family-specific fine-tuning."

bmDCA training:In figure 3 middle left panel (comparing MSA covariance versus bmDCA-generated covariance) shows a fairly small regression line slope (0.67). This suggests too large a regularization on couplings and/or an insufficient number of iterations. Have you checked that the algorithm successfully converged, or was the maximum number of iterations reached? What about using a smaller regularization strength (two values were used in Russ et al. Science 2020, one small and one large)? I know that parameter tuning is a weakness of bmDCA, but the proposed method also has free parameters and it is not clear whether they are universal for all protein lengths. This could also explain the discrepancy in the topology of the sequence distribution shown in Figure 5 (weak couplings cannot create "ferromagnetic" multimodal distributions).

We did our best to reproduce the published bmDCA results, and to train bmDCA as in published papers. Our baseline fitting method employs the default parameters of [Figliuzzi et al. 2018] and uses the exact same convergence criteria, and our sampling method employs the equilibration time determined there. We chose to employ the default parameters to allow for direct comparison with the results of [Figliuzzi et al. 2018]. Below, we show quantitative comparisons of the fitting performance of two-body correlations between our bmDCA results and published ones.

More generally, we agree with the reviewer about the importance of regularization strength and temperature in bmDCA, as reported in [Russ et al. 2020]. Throughout our revised manuscript, we now present results with parameter values used in [Russ et al. 2020], in addition to our results with default parameters. We explain this important point in more detail in our response to the next reviewer question.

Details about bmDCA fitting quality:

We checked convergence and we double-checked that our results are consistent with these published results and those of [Trinquier et al. 2021]. In particular, the Pearson correlation values between the pairwise correlations in the natural and bmDCA-generated data are consistent with those reported in [Figliuzzi et al. 2018] and [Trinquier et al. 2021]:

**Author response table 1. sa2table1:** 

Family	Pearson (Figliuzzi/Trinquier)	Pearson (ours)
PF00004	0.95/-	0.98
PF00005	0.95/-	0.96
PF00041	0.97/-	0.96
PF00072	0.98/0.93	0.96
PF00076	0.98/0.97	0.95
PF00096	0.99/-	0.93
PF00153	0.97/-	0.92
PF00595	-/0.97	0.93
PF01535	0.99/-	0.98
PF02518	0.97/-	0.97
PF07679	0.95/-	0.96

As can be observed from this table, our results are similar to the published ones, and the range of differences between our results and published ones appears comparable to that between the results of two papers by the same group.

Regarding the slope of 0.67 mentioned by the reviewer for PF00153 (now shown in Figure 4—figure supplement 3), unfortunately, we cannot compare it directly to previous results, because the value of this slope was not reported in previous works. However, there are 3 other families for which this slope is explicitly reported in [Trinquier et al. 2021], and here is the comparison in this case:

**Author response table 2. sa2table2:** 

Family	Slope (Trinquier)	Slope (ours)
PF00072	0.74	0.88
PF00076	0.92	0.82
PF00595	0.88	0.75

From these three examples, our slopes do not seem further from one than in [Trinquier et al. 2021].

Evaluation of sequence quality:I feel that the comparison of Figure 1 is a bit unfair. On the one hand, the proposed MSA-Transformer sampling method generates samples from the T=0 conditional distribution (according to step 3, the bottom of p9), whereas the bmDCA generates samples from the T=1 distribution. It is known that samples from the T=1 distribution have worse statistical energy scores than natural sequences due to the regularization of fields/couplings and that this must be adequately compensated by low-temperature sampling (Russ et al. Science 2020). The authors argue p13 that low-temperature sampling results in distortion of the first and second-order moments of the data, but I think this is not a practical problem for protein design (by definition, an optimal design protocol will discard all suboptimal amino acids for a given position and only retain the most favorable ones).

We thank both reviewers for raising this very interesting point. We have now performed a comprehensive comparison of our MSA-Transformer–generated data not only to bmDCA-generated data at sampling temperature T=1 but also at lower sampling temperatures. We considered the two temperature values chosen in [Russ et al. 2020], namely T=0.33 and T=0.66. For completeness, we also considered the two values of regularization strength λ from [Russ et al. 2020] for these three temperatures, in the case of family PF00072, as reported in Table 4. Given the relatively small impact of λ observed there, we kept only one value of λ for each value of T in the rest of our manuscript – namely, λ=0.01 for T=1 to match the parameters in [Figliuzzi et al. 2018], and λ=0.001 for T=0.33 and T=0.66 as it gave slightly better scores in Table 4. Note that for our additional study of small protein families (see below), we employed λ=0.01 throughout because it is better suited to small families.

In particular, to make Figure 1 fairer, as per the reviewer's remark, we now include results obtained for bmDCA at λ=0.001 and T=0.33 in this figure. For completeness, we also include them in all other figures of the revised manuscript. Results for T=0.66 were quite similar to those obtained for T=0.33, and we also report them in Table 4 and Figure 5—figure supplements 1-2 for completeness.

Our general findings, which are discussed in the revised manuscript, are that decreasing T indeed improves the scores of bmDCA-generated sequences. However, the main improvement regards statistical energy (as expected from lowering T), while the improvements of other scores (HMMER score, and, more importantly, structural scores) are more modest. Even using T=0.33 for bmDCA, our MSA-Transformer–generated sequences have similar or better scores compared to bmDCA-generated sequences, apart from statistical energy (see Figure 1 and Table 1 and 1c). Moreover, we find that decreasing T with bmDCA substantially decreases MSA diversity, while MSA-Transformer–generated sequences do not suffer from such an issue (see Figure 1—figure supplement 1). In fact, at low T, bmDCA concentrates on local minima of the statistical energy landscape (see Figures 2, 5 and Figure 5—figure supplements 1-2), resulting in low diversity.

Overall, these new results confirm that our procedure for generating sequences using MSA Transformer is promising, featuring scores at least comparable with low-temperature bmDCA sequences, and high diversity.

The comparison of AlphaFold confidence scores between MSAtransformer-generated and natural sequences is interesting but the results are not discussed at all in the main text. Is there a statistically significant difference between the distributions? Moreover, it is not shown here whether the predicted structures are similar to the native ones. Please provide a fold similarity metric such as backbone RMSD values.

We thank the reviewer for these important remarks. We now present an analysis of the statistical significance of the difference between the distributions of all scores in the various generated datasets in Table 1, using the Kolmogorov-Smirnov test, and we also compare the median and standard deviation of all scores in the natural and generated datasets in Table 2. We revised the discussion in the main text accordingly.

In addition, we agree that checking whether the predicted structures are similar to the native ones is an important test that goes beyond the AlphaFold pLDDT. We therefore added an additional score throughout our manuscript, which is the RMSD between a reference experimental structure of the family (see Table 5) and the AlphaFold structure predicted for each sequence studied. The results from the RMSD analysis corroborate those obtained with pLDDT and show that predicted structures are indeed similar to the native ones. These results are now discussed in the main text. We believe that this point strengthens our conclusions, and we thank the reviewer for suggesting this analysis.

Evaluation of sample diversity:The authors convincingly show that sequences generated by MSAtransformer differ substantially from natural sequences. However, given that the MSA-Transformer samples are essentially obtained by zero-temperature dynamics, it is important to assess how diverse are the generated sequences. This is partially addressed in Figure S5, but consider using simpler metrics. Can the authors provide basic estimates of sample diversity? (e.g., effective number of sequences as a function of the number of samples).

We thank the reviewer for asking this interesting question. We added a new supplementary figure, Figure 1—figure supplement 1, to address this point. In this figure, we show the effective number of sequences Meff for the MSAs we generated, versus the similarity threshold employed to define Meff. We find that the sequence diversity of MSA-Transformer–generated MSAs is slightly smaller than that of the natural MSAs, but remains of the same order of magnitude. Therefore, the method we propose to generate sequences using MSA Transformer preserves diversity to a large extent, despite using zero-temperature dynamics. This is probably enabled by the fact that we start from an MSA and not from a single sequence, and that MSA

Transformer uses the whole MSA as context.

Conversely, Figure 1—figure supplement 1 shows that low-temperature bmDCA sampling leads to substantially reduced Meff values, consistently with the idea that only the local minima of the energy landscape are then sampled. More precisely, we observe a regular increase of Meff with the similarity threshold for MSA-Transformer–generated data as well as for natural data, while the increase is more stepwise for low-temperature bmDCA.

Reviewer #2 (Recommendations for the authors):I have a series of questions that if resolved could help, in my opinion, to create an improved version of this interesting study.1. The manuscript describes how MSA Transformers could lead to metrics that are "even better" than the natural sequences. I feel that this statement is a bit misleading as the natural sequences are a baseline to compare. There are no better properties than those of the natural sequences because that is what we observe in nature. I would suggest removing those statements as they might produce confusion. On the other hand, if the authors would provide examples of sequences that are "better" in terms of function, stability, or any other quantifiable metric, then I would agree that these statements would make more sense.

We agree with the reviewer that natural sequences are the reference here. We changed the wording and we no longer claim that generated sequences have "better scores", in order to avoid any confusion about this.

2. The authors mention methods that can be trained in an unsupervised way to predict structure based on a single sequence. I wonder if this statement can be clarified, as it is obvious that those methods use several sequences as training. Are these methodologies based on physical principles instead of learning from sequences?

We agree that this statement was a bit misleading, and we thank the reviewer for pointing it out. We rephrased it to clarify that these models, which are trained on a large database of sequences, are then able to predict the structure taking as input either a single sequence (ESM1b and now ESM2) or an MSA (MSA Transformer). The corresponding sentence now reads:

“These pre-trained models are able to predict structure in an unsupervised way [21], either taking as input a single sequence [20] or an MSA [25], potentially by transferring knowledge from their large training set [26, 27].”

3. It is not clear to me what is the importance of the correlation between HMMER and Hamming distance, the authors should provide more intuition on why this is a relevant metric. Since those correlations are quite low for both models I am not sure if it contributes to the analysis in a meaningful way.

We thank the reviewer for raising this important point. One may have feared that generated sequences that are most similar to natural ones might always have higher scores, which would show that the model has difficulty to generalize and may be overfitting. The fact that HMMER scores tend to be positively correlated with Hamming distances for sequences generated using our method based on MSA Transformer is one of several pieces of evidence that this is not the case. However, we agree that our detailed discussion of the correlations between HMMER scores and Hamming distances gave too much emphasis to this point. Therefore, we have now strongly reduced this discussion.

4. Are the contact maps created from the generated MSA of Transformer better than those of bmDCA? My understanding is that the value of p=0.1 optimizes contact map accuracy, but how are those compared against DCA maps?

It was shown in the original MSA Transformer paper, [Rao et al. 2021], that MSA Transformer outperforms DCA at unsupervised contact prediction on natural data. In addition, in Figure 7—figure supplements 1 to 4, we show that for small values of the masking probability p (e.g. p=0.1), the contact maps inferred by MSA Transformer from our MSAs generated using our iterative masking procedure based on MSA Transformer reproduce quite well those of natural MSAs of the same protein family. The goal here was to check that MSA Transformer was remaining within the protein family of focus when using our generation method. We now clarified this point.

In addition, for the protein family PF00072, we now also used fully synthetic MSAs (either generated using our method based on MSA Transformer or generated by bmDCA) as input to AlphaFold (Table 4). We find that structural scores (both pLDDT and RMSD) are fully in line with those of natural data, both for our method based on MSA Transformer and for low-temperature bmDCA, while bmDCA at T=1 gives poorer results.

5. Haldane et al. (https://doi.org/10.1016/j.bpj.2017.10.028) has proposed how pairwise statistics seem to be enough to recapitulate higher order mutational patterns in natural sequences. Could the authors comment on this, and mention if there is a substantial advantage in capturing 3-body statistics in the process of protein design or generative modeling?

We thank the reviewer for pointing out this interesting study that we now cite. We took inspiration from it, and from the later study [McGee et al. 2021] by some of the same authors, and investigated higher-order statistics in natural and synthetic MSAs (beyond 2- and 3-body ones that were already studied in the first version of our manuscript). Our new Figure 4 shows the r20 score that quantifies the similarity of statistics at various orders between two datasets. It also presents as reference an assumption-free null model, namely the r20 score between two subsets of each natural MSA. As now mentioned in the main text,

"Figure 4 shows that bmDCA with default parameters is most often the best method at reproducing lower-order statistics, but that MSA Transformer is the best at reproducing higher-order statistics, in all families considered."

While pairwise models are extremely successful at modeling the statistics of protein MSAs and at making predictions e.g. about their structure, interactions, and mutational effects, this does not necessarily imply that higher-order statistics are always fully captured by these models. Besides, the training of these models is difficult and subject to finite size issues, as discussed e.g. in [McGee et al. 2021]. We note that in the context of neuroscience [Meshulam et al. 2021], a better agreement is obtained on 3-body correlations using pairwise maximum entropy models.

We now mention this point: "Thus, bmDCA is trained to reproduce these frequencies, and achieves these objectives quite well, although the comparison to the null model in Figure 4—figure supplement 2 and Figure 4—figure supplement 3 hints that further improvements remain possible, see [51]." Either way, here, we take these results as a pragmatic indication that lower-order statistics are better reproduced by bmDCA while higher-order statistics are better reproduced by our method based on MSA Transformer.

6. It would be interesting to see how Alphafold would perform if some of these generated MSAs are used as input. I don't think it is needed to test it with all the families, but it would be an interesting experiment for a single family.

We thank the reviewer for this interesting suggestion. As mentioned above, for the protein family PF00072, we now used fully synthetic MSAs (either generated using our method based on MSA Transformer or generated by bmDCA) as input to AlphaFold (Table 4). We find that structural scores (both pLDDT and RMSD) are then very similar to those obtained using natural context (which is our usual procedure). They are also fully in line with those of natural data for our method based on MSA Transformer, illustrating the robustness of the good structural scores of our synthetic sequences. We have added the sentence:

“In addition, for the protein family PF00072, we also used fully synthetic MSAs as input to AlphaFold. Structural scores are then very similar to those obtained using natural context (see Table 4)”.

7. The families used in this study appear to have enough statistics to perform well, this is reasonable, however, for the sake of comparison, it would be interesting to see how the MSA Transformer would compare against bmDCA for a family with just a few sequence members. Are the trends the same? Or do we see a change in performance between the two generative methodologies?

We thank the reviewer for asking this extremely interesting question. We now present new results for smaller protein families, listed in Table 6, whose shallow MSAs make it more challenging to accurately fit Potts models. Our results are shown in the new Figure 3, and discussed in the main text, in a new section titled "Sequence generation by the iterative masking procedure is successful for small protein families" at the end of Results. As mentioned there,

“We observe that MSA-Transformer–generated sequences have similar HMMER scores and structural scores to natural sequences. MSA-Transformer–generated sequences also generally have better HMMER scores and structural scores than those generated by bmDCA with default parameters. While low-temperature bmDCA yields better statistical energy scores (as expected), and also gives HMMER scores and structural scores comparable to natural sequences, it in fact generates sequences that are almost exact copies of natural ones. By contrast, MSA Transformer produces sequences that are quite different from natural ones, and have very good scores. Thus, our method based on MSA Transformer is particularly promising in the tricky case of small protein families.”

This analysis shows that our method not only performs as well as bmDCA for large families, but also has a broader scope, as it is less limited by MSA depth than bmDCA. We believe that this makes our manuscript stronger, and we thank the reviewer again for suggesting this very relevant additional study.